# The postsynaptic t-SNARE Syntaxin 4 controls traffic of Neuroligin 1 and Synaptotagmin 4 to regulate retrograde signaling

Kathryn P Harris[1,2,3]*, Yao V Zhang[1,2,3], Zachary D Piccioli[1,2,3], Norbert Perrimon[4,5], J Troy Littleton[1,2,3]

[1]The Picower Institute for Learning and Memory, Massachusetts Institute of Technology, Cambridge, United States; [2]Department of Biology, Massachusetts Institute of Technology, Cambridge, United States; [3]Department of Brain and Cognitive Sciences, Massachusetts Institute of Technology, Cambridge, United States; [4]Department of Genetics, Harvard Medical School, Boston, United States; [5]Howard Hughes Medical Institute, Harvard Medical School, Boston, United States

**Abstract** Postsynaptic cells can induce synaptic plasticity through the release of activity-dependent retrograde signals. We previously described a $Ca^{2+}$-dependent retrograde signaling pathway mediated by postsynaptic Synaptotagmin 4 (Syt4). To identify proteins involved in postsynaptic exocytosis, we conducted a screen for candidates that disrupted trafficking of a pHluorin-tagged Syt4 at *Drosophila* neuromuscular junctions (NMJs). Here we characterize one candidate, the postsynaptic t-SNARE Syntaxin 4 (Syx4). Analysis of *Syx4* mutants reveals that Syx4 mediates retrograde signaling, modulating the membrane levels of Syt4 and the transsynaptic adhesion protein Neuroligin 1 (Nlg1). Syx4-dependent trafficking regulates synaptic development, including controlling synaptic bouton number and the ability to bud new varicosities in response to acute neuronal stimulation. Genetic interaction experiments demonstrate *Syx4, Syt4*, and *Nlg1* regulate synaptic growth and plasticity through both shared and parallel signaling pathways. Our findings suggest a conserved postsynaptic SNARE machinery controls multiple aspects of retrograde signaling and cargo trafficking within the postsynaptic compartment.

*For correspondence: kpharris@mit.edu

**Competing interests:** The authors declare that no competing interests exist.

## Introduction

Synaptic connections form and mature through signaling events in both pre- and postsynaptic cells. The release of signaling molecules into the synaptic cleft depends on SNARE proteins that drive membrane fusion. This machinery is well understood for neurotransmitter release from the presynaptic cell: in response to an action potential, a v-SNARE in the synaptic vesicle membrane (Synpatobrevin/VAMP) engages t-SNARES in the presynaptic membrane (Syx1 and SNAP-25), forming a four-helix structure that brings the membranes into close proximity and initiates fusion (*Jahn and Scheller, 2006*; *Südhof and Rothman, 2009*). Although SNARE-dependent fusion drives membrane dynamics in all cell types, it is specialized in the presynaptic terminal to be $Ca^{2+}$-dependent, employing $Ca^{2+}$ sensors like Synaptotagmin 1 (Syt1) to link synaptic vesicle fusion to $Ca^{2+}$ influx following an action potential.

The postsynaptic cell also exhibits activity-dependent exocytosis. Altering the composition of the postsynaptic membrane, including regulated trafficking of neurotransmitter receptors, is an important plastic response to neural activity (*Chater and Goda, 2014*). The postsynaptic cell also releases

**eLife digest** Synapses are connections that allow a neuron to communicate with a neighboring cell (often another neuron). When an electrical impulse traveling down the "presynaptic" neuron reaches the synapse, it causes the neuron to release molecules called neurotransmitters. These molecules then bind to receptors on the surface of the other "postsynaptic" cell and cause that cell to respond in a particular way.

Communication between the two cells at the synapse can also go in the opposite direction, with the postsynaptic cell signaling to the presynaptic cell. Such "retrograde" signals typically regulate the properties of the synaptic connection, such as changing the strength or shape of the synapse, or altering which proteins are present there.

While a lot is known about how a presynaptic neuron communicates with the postsynaptic cell, not as much is known about how retrograde signals are regulated. Harris et al. therefore set out to identify and characterize new factors that control retrograde signaling, and started by producing a list of likely candidate molecules. These candidates were then screened by removing them one at a time from the synapses between motor neurons and muscle cells in fruit flies and observing the effect this had on a molecule called Synaptotagmin 4.

Synaptotagmin 4 is normally found at the membrane of the postsynaptic cell. Harris et al. found that removing one candidate molecule, called Syntaxin 4, from the postsynaptic cell reduced the amount of Synaptotagmin 4 at the membrane. Further investigation showed that Syntaxin 4 also helps to deliver a protein called Neuroligin 1 to the postsynaptic membrane, which is important for organizing the synapse.

By identifying Syntaxin 4 as a new regulator of retrograde signaling, Harris et al. open up several avenues of investigation that could reveal more about the mechanisms that influence how synapses work.

retrograde signals into the synaptic cleft to modulate synaptic growth and function. These retrograde messengers include lipid-derived molecules like endocannabinoids (*Ohno-Shosaku and Kano, 2014*), gases like nitric oxide (*Hardingham et al., 2013*), neurotransmitters (*Koch and Magnusson, 2009*; *Regehr et al., 2009*), neurotrophins (*Zweifel et al., 2005*), and other signaling factors like TGF-β and Wnt (*Poon et al., 2013*; *Salinas, 2005*; *Sanyal et al., 2004*; *Speese and Budnik, 2007*). Adhesion complexes that provide direct contacts across the synaptic cleft also participate in retrograde signaling (*Futai et al., 2007*; *Gottmann, 2008*; *Hu et al., 2012*; *Mozer and Sandstrom, 2012*; *Peixoto et al., 2012*; *Vitureira et al., 2012*).

Although retrograde signaling is a key modulator of synaptic function, little is known about how postsynaptic exocytosis is regulated and coordinated. Components of a postsynaptic SNARE complex have been recently identified in mammalian dendrites. The t-SNAREs Syntaxin 3 (Stx3) and SNAP-47 are required for regulated AMPA receptor exocytosis during long term potentiation, while the v-SNARE synaptobrevin-2 regulates both activity-dependent and constitutive AMPAR trafficking (*Jurado et al., 2013*). Stx4 has also been implicated in activity-dependent AMPAR exocytosis (*Kennedy et al., 2010*). In *Drosophila*, a $Ca^{2+}$-dependent retrograde signaling pathway relies on the postsynaptic $Ca^{2+}$ sensor Syt4. Syt4 vesicles fuse with the postsynaptic membrane in an activity-dependent fashion (*Yoshihara et al., 2005*), and loss of *Syt4* leads to abnormal development and function of the NMJ. *Syt4* null animals have smaller synaptic arbors, indicating a defect in synaptic growth, and also fail to exhibit several forms of synaptic plasticity seen in control animals, including robust enhancement of presynaptic release in response to high frequency stimulation, and rapid budding of synaptic boutons in response to strong neuronal stimulation (*Barber et al., 2009*; *Korkut et al., 2013*; *Piccioli and Littleton, 2014*; *Yoshihara et al., 2005*). However, a detailed understanding of how the postsynaptic cell regulates constitutive and activity-dependent signaling of multiple retrograde pathways is lacking. In addition to exocytosis, it is likely that many cellular processes including vesicle trafficking and polarized transport of protein and transcript are specialized to facilitate postsynaptic signaling. Identifying such regulatory mechanisms is crucial for understanding synaptic development and function.

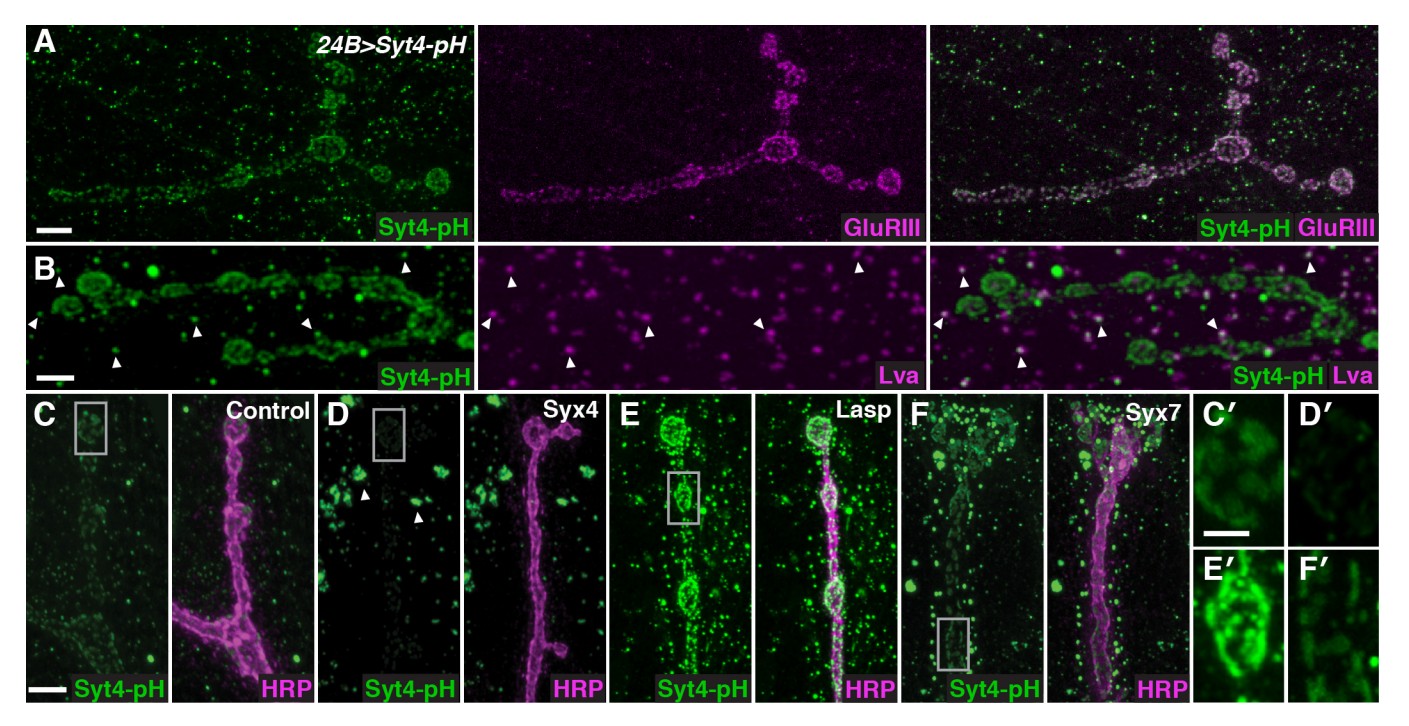

**Figure 1.** A candidate RNAi screen for regulators of postsynaptic exocytosis. (A,B) Representative images of Syt4-pH expressed with the postsynaptic muscle driver 24B-GAL4. Syt4-pH (green) accumulates in postsynaptic membrane that also contains domains of GluRIII (magenta) (A). Syt4-pH also decorates numerous cytoplasmic puncta, many of which overlap with the Golgi marker Lva (magenta), arrowheads (B). (C–F) Examples of candidate RNAis affecting Syt4-pH localization: control (C); *Syx4-RNAi* reduces Syt4-pH at the membrane and causes a redistribution to prominent cytoplasmic puncta, arrowheads (D); *Lasp-RNAi* increases Syt4-pH at the membrane (E); and *Syx7-RNAi* causes a redistribution of Syt4-pH puncta around the NMJ without affecting the intensity at the membrane (F). (C′–F′) Close-ups of C–F. Scale bars = 7 μm (A), 5 μm (B–F), 2 μm (C′–F′).

The following figure supplements are available for figure 1:

**Figure supplement 1.** Both Syt4-GFP CRISPR knock-in and overexpression of Syt4-pH can replace endogenous Syt4 with respect to synaptic architecture and plasticity.

**Figure supplement 2.** pHluorin is quenched in live but not fixed preparations.

We conducted a candidate-based transgenic RNAi screen to identify regulators of postsynaptic exocytosis at the *Drosophila* NMJ, a model for studying glutamatergic synapse growth and plasticity (*Harris and Littleton, 2015*). Using a fluorescently tagged form of the postsynaptic $Ca^{2+}$ sensor Syt4, we screened for candidate gene products that disrupted the localization of Syt4 at the postsynaptic membrane. Here we describe our characterization of one candidate from this screen, *Syntaxin 4 (Syx4)*. *Drosophila* Syx4 is the sole homolog of the mammalian Stx 3/4 family of plasma membrane t-SNAREs that also includes Syntaxin 1 (*Littleton, 2000*). The mammalian Stx3 and Stx4 homologs regulate activity-dependent AMPA receptor trafficking in mammalian neurons (*Jurado et al., 2013*; *Kennedy et al., 2010*), while Stx4 also participates in regulated secretory events in several other mammalian cell types, including insulin-stimulated delivery of the glucose transporter to the plasma membrane in adipocytes and glucose-stimulated insulin secretion from pancreatic beta cells (reviewed by *Jewell et al., 2010*). Our results demonstrate that the *Drosophila* Syx4 homolog is essential for retrograde signaling, regulating the membrane delivery of both Syt4 and Neuroligin (Nlg1), a transsynaptic adhesion protein that plays important roles in synapse formation and function, and is linked to autism spectrum disorder (ASD) (*Bang and Owczarek, 2013*; *Bottos et al., 2011*; *Südhof, 2008*). Through genetic interaction experiments, we define functions of the Syx4, Syt4, and Nlg1 pathway in regulating multiple aspects of synaptic growth and plasticity within the postsynaptic compartment.

## Results

### A candidate RNAi screen for regulators of postsynaptic exocytosis

To identify regulators of Syt4 trafficking, we conducted a candidate-based RNAi screen at the NMJ. Our screening approach employed transgenic animals expressing Syt4 tagged with pHluorin, a pH-sensitive variant of GFP under the control of the UAS promoter (*UAS-Syt4-pH*). When expressed with the muscle driver *24B-GAL4*, Syt4-pH protein decorates the postsynaptic membrane of the NMJ, overlapping with glutamate receptor (GluR) fields opposite active zones (AZs) (*Yoshihara et al., 2005*; *Figure 1A*). Syt4-pH is also found in numerous vesicular structures throughout the muscle, many of which overlap with the Golgi marker Lava lamp (Lva) (*Figure 1B*). Postsynaptic expression of Syt4-pH rescues synaptic phenotypes previously reported in *Syt4* null animals (*Figure 1—figure supplement 1*), including a decrease in the number of synaptic boutons and a decrease in the ability to grow new boutons ("ghost boutons", or GBs) in response to strong neuronal stimulation (*Barber et al., 2009*; *Korkut et al., 2013*; *Piccioli and Littleton, 2014*; *Yoshihara et al., 2005*). Thus, Syt4-pH is functional at the NMJ.

Candidate *UAS-RNAi* constructs were co-expressed with *UAS-Syt4-pH* in muscle, and animals were examined for changes in Syt4-pH distribution. We looked for changes in Syt4-pH intensity at the postsynaptic membrane (defined as discreet Syt4-pH fields adjacent to the neuronal membrane), or other changes in the distribution, size or intensity of Syt4-pH-positive vesicular structures. Resolution of Syt4-pH localization was best achieved following tissue fixation, which is expected to interfere with the pH sensitivity of the pHluorin tag. Indeed, treatment of fixed samples with a low pH (5.5)

**Table 1.** RNAis that alter the localization of Syt4-pH Candidate gene products are listed, along with the predicted gene function, the specific effect on Syt4-pH, and the RNAi constructs tested. RNAi lines were obtained from the Transgenic RNAi Project (TRiP) at Harvard Medical School (*Perkins et al., 2015*)[a] or the Vienna *Drosophila* RNAi Center (*Dietzl et al., 2007*)[b].

| Gene product | CG | Function | Syt4-pH distribution | RNAis |
|---|---|---|---|---|
| Syntaxin 4 | CG2715 | t-SNARE | Reduced intensity at NMJ, large clusters in cytoplasm | JF01714[a] V32413[b] |
| Syntaxin 6 | CG7736 | t-SNARE | Reduced intensity at NMJ, large clusters in cytoplasm | V1579[b] V1501[b] |
| Syntaxin 18 (Gtaxin) | CG13626 | t-SNARE | Reduced intensity at NMJ, large clusters in cytoplasm | JF02263[a] |
| MyoV | CG2146 | Dilute class unconventional myosin | Reduced intensity at NMJ, large clusters in cytoplasm | JF03035[a] V16902[b] |
| Actin-related protein 2/3 complex, subunit 3A | CG4560 | Arp2/3 complex-mediated actin nucleation | Reduced intensity and size at NMJ, smaller cytoplasmic puncta | JF02370[a] |
| Gdi | CG4422 | Rab GDP-dissociation inhibitor | Reduced intensity at NMJ, large clusters in cytoplasm | JF02617[a] V26537[b] |
| Rabex-5 | CG9139 | Rab5 guanyl-nucleotide exchange factor activity | Reduced intensity at NMJ, large clusters in cytoplasm | JF02521[a] |
| Lasp | CG3849 | Actin binding | Increased intensity at NMJ | JF02075[a] |
| Neuroglian | CG1634 | Cell adhesion; axon guidance; synapse organization | Increased intensity at NMJ | JF03151[a] V27201[b] |
| Contactin | CG1084 | Cell adhesion | Increased intensity at NMJ | HM05134[a] HMS00186[a] |
| Syntaxin 7 | CG5081 | t-SNARE, early endosomal regulation | Intensity at NMJ normal, many small bright puncta cluster adjacent to NMJ | JF02436[a] V5413[b] |
| Dynamin associated protein 160 | CG1099 | Synaptic vesicle endocytosis; cell polarity | Intensity at NMJ normal, many small bright puncta cluster adjacent to NMJ | JF01918[a] V16158[b] |
| Adaptor Protein complex 2, σ subunit | CG6056 | Endocytosis | Intensity at NMJ normal, many small bright puncta cluster adjacent to NMJ | JF02631[a] |
| Adaptor Protein complex 2, α subunit | CG4260 | Endocytosis | Intensity at NMJ normal, many small bright puncta cluster adjacent to NMJ | HMS00653[a] |
| β-spectrin | CG5870 | Cytoskeleton; synapse organization | Irregular size and spacing at NMJ | HMS01746[a] V42053[b] |

adjusted buffer did not affect our detection of Syt4-pH in fixed tissue, compared to a dramatic quenching of fluorescence that was observed in a live preparation (*Figure 1—figure supplement 2*). Thus, we interpret the Syt4-pH localization pattern in our fixed-tissue assay as non-pH-dependent.

We assembled a candidate list of gene products resident at synapses and/or involved in membrane trafficking (*Supplementary file 1*) using the following criteria: 1) *Drosophila* orthologs of proteins identified in proteomics studies of mouse and rat brain synaptic membranes (*Abul-Husn et al., 2009*; *Li et al., 2007c*); 2) candidate genes identified in a *Drosophila* screen for transposon insertions affecting glutamate receptor expression or localization (*Liebl and Featherstone, 2005*); and 3) known regulators of membrane trafficking (eg Rabs, SNARE proteins, Vps proteins). Transgenic RNAi lines were obtained from the Transgenic RNAi Project (TRiP) at Harvard Medical School (*Perkins et al., 2015*) or the Vienna *Drosophila* RNAi Center (*Dietzl et al., 2007*). For 190 candidates that had no RNAi stock already available, transgenic RNAi stocks were generated by the TRiP at Harvard Medical School; these stocks are currently available from the Bloomington stock center.

Among the 442 lines screened, 15 candidates were identified with abnormal Syt4-pH distribution (*Table 1*). These candidates fell into three qualitatively distinct categories based on the intensity of Syt4-pH at the postsynaptic membrane: decreased intensity of Syt4-pH (7/15, *Figure 1D,D′*, eg. *Syx4-RNAi*), increased intensity of Syt4-pH (3/15, *Figure 1E,E′*, eg. *Lasp-RNAi*), and changes in the distribution of Syt4-pH-positive vesicles with otherwise normal intensity (5/15, *Figure 1F,F′*, eg. *Syx7-RNAi*). Two candidates had phenotypes consistent with previously published studies, supporting the efficacy of the screen: knockdown of *Syx18/Gtaxin* dramatically reduced Syt4 delivery, consistent with a role in postsynaptic membrane addition (*Gorczyca et al., 2007*), and knockdown of *β-spectrin* resulted in changes in the spacing of Syt4-pH domains, consistent with a role for the spectrin cytoskeleton in AZ/GluR spacing (*Pielage et al., 2006*). We chose to focus our analysis on one candidate, the plasma membrane t-SNARE Syx4. Knockdown of *Syx4* produced a decrease in the intensity of Syt4-pH at the postsynaptic membrane, along with large accumulations of Syt4-pH in the cytoplasm (*Figure 1D,D′*), suggesting that Syx4 may regulate membrane levels of Syt4 and modulate Syt4-dependent signaling mechanisms.

## Syntaxin 4 is enriched postsynaptically at the NMJ

To investigate the function of *Syx4*, we created mutant alleles by mobilizing a transposable P-element located in the 5'-UTR of the *Syx4* locus (*Figure 2A*). *Syx4* encodes a protein with a large N-terminal domain, a SNARE domain, and a C-terminal transmembrane domain (*Figure 2B*). Two Syx4 proteins (Syx4A and Syx4B) are predicted from genome analysis, resulting in a longer (A) or shorter (B) N-terminus. RT-PCR analysis indicated that both of these isoforms are expressed in *Drosophila* larvae (*Figure 2—figure supplement 1*).

We isolated three alleles that delete parts of the *Syx4* coding region. *Syx4³⁹* carries a deletion from the 5'-UTR to the first intron, removing the first exon and the start site for the *Syx4A* isoform. *Syx4⁴⁸* carries a deletion from the 5'-UTR to the second intron, removing the first two exons and both predicted start sites. Finally, *Syx4⁷³* carries a deletion from the 5'-UTR through the entire coding region of the gene. Several lines of evidence discussed below indicate *Syx4⁷³* is a null allele. A precise excision with no deletion was also generated, and was used as a genetic background control.

To examine the subcellular distribution of Syx4, we generated polyclonal antisera against purified Syx4A protein. Syx4 is expressed in the muscle at the NMJ and is enriched postsynaptically, as revealed by co-staining with an antibody against HRP to highlight the presynaptic membrane (*Figure 2C*). This staining is absent in *Syx4⁷³* mutants (*Figure 2D*), consistent with this allele removing the entire coding region of the gene. We also produced *UAS-Syx4* and *UAS-RFP-Syx4* constructs for both protein isoforms, in order to overexpress untagged or tagged Syx4 at the NMJ. Expression of RFP-Syx4A (*Figure 2E*) or RFP-Syx4B (data not shown) with the postsynaptic muscle driver *24B-GAL4* showed a similar distribution to the endogenous protein. Thus, *Syx4* is expressed in the postsynaptic cell and accumulates at the synaptic membrane.

We also tested the *Syx4⁷³* allele for its effect on the distribution of Syt4-pH. Similar to our RNAi knockdown results, *Syx4* mutants exhibited a decrease in Syt4-pH intensity at the NMJ, accompanied by a redistribution of Syt4-pH to large cytoplasmic accumulations (*Figure 2F,G,F′,G′*).

To test whether Syx4 regulates the localization of endogenous Syt4, we used CRISPR/CAS9 to generate a C-terminal GFP knock-in line, *Syt4^{GFP-2M}*. We produced a transgenic stock expressing

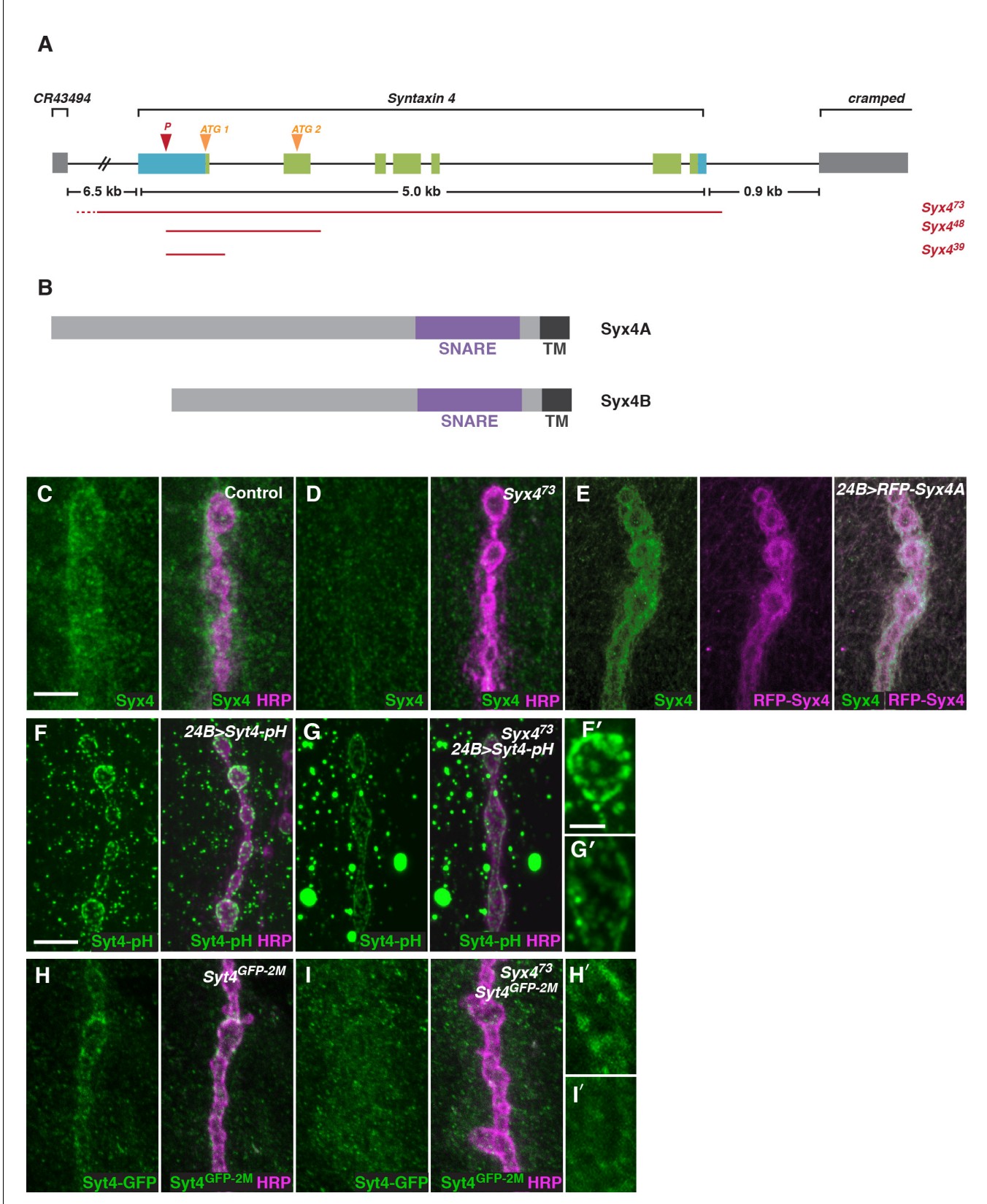

**Figure 2.** Syntaxin 4 is a postsynaptic plasma membrane SNARE. (**A**) *Syx4* genomic region. Coding exons are indicated in green while non-coding exons are in blue. Two predicted start sites (ATG) are indicated in orange. The location of the P-element used for mutagenesis (P) is indicated in red. *Figure 2 continued on next page*

*Figure 2 continued*

Three alleles of *Syx4* were isolated. Deleted regions are indicated in red. Solid lines indicate regions known to be deleted from PCR analysis and sequencing, while dotted lines indicate regions within which breakpoints have been mapped. (**B**) *Syx4* encodes a protein with an N-terminal domain, a SNARE domain and a C-terminal transmembrane domain. There are two predicted isoforms that differ in the size of the N-terminal domain. (**C,D**) Representative images of NMJs stained for Syx4 (green) and the neuronal membrane marker HRP (magenta). Syx4 staining at the synapse in precise excision control animals (**C**) is absent in *Syx4^73^* mutant animals (**D**). (**E**) Representative image from an animal stained for Syx4 (green) and expressing *RFP-Syx4* (magenta) with *24B-GAL4*. (**F,G**) Representative images from animals expressing *Syt-pH* with *24B-GAL4* in a control (**F**) or *Syx4^73^* (**G**) background. Syt4-pH is reduced at the postsynaptic membrane and redistributed to large cytoplasmic accumulations in *Syx4^73^* mutants. (**F',G'**) Close-ups of F and G. (**H,I**) Representative images from *Syt4^GFP-2M^* knock-in animals in a control (**H**) or *Syx4^73^* (**I**) background. Synaptic localization of Syt4^GFP-2M^ is reduced in *Syx4^73^* mutants. (**H',I'**) Close-ups of H and I. Scale bars = 5 μm (**C–I**), 2 μm (**F',G',H',I'**).

The following figure supplement is available for figure 2:

**Figure supplement 1.** RT-PCR analysis of Syntaxin 4.

custom guide RNAs (gRNAs) targeting the *Syt4* locus. These animals were crossed to transgenic flies expressing germline-specific Cas9, and embryos from the cross were injected with a donor plasmid for homology-directed repair to insert GFP into the *Syt4* genomic locus (*Gokcezade et al., 2014*; *Kondo and Ueda, 2013*; *Port et al., 2014*). The *Syt4^GFP-2M^* line is homozygous viable and fertile, and does not exhibit synaptic defects that have been previously described in animals lacking *Syt4* (*Figure 1—figure supplement 1*), indicating that Syt4^GFP-2M^ protein is functional.

Syt4^GFP-2M^ shows synaptic localization at the NMJ (*Figure 2H,H'*), and this localization is lost in the *Syx4* null mutant background (*Figure 2I,I'*). These findings indicate that the distribution of Syt4-pH reported in our screen is recapitulated by endogenous Syt4 protein. As we do not observe large cytoplasmic accumulations of Syt4^GFP-2M^ in *Syx4* mutants, this feature likely results from overexpression of Syt4-pH protein using the *24B-GAL4* driver.

## Syntaxin 4 is required postsynaptically to regulate synaptic growth

Because *Syx4* mutants have a defect in Syt4 localization, we investigated whether *Syx4* impacts synaptic development in a similar manner to *Syt4*. *Syt4* null mutants show abnormal development and function of the NMJ, including a decrease in the number of synaptic boutons, and a failure to express several forms of synaptic plasticity (*Barber et al., 2009*; *Korkut et al., 2013*; *Piccioli and Littleton, 2014*; *Yoshihara et al., 2005*). We first quantified the number of boutons per NMJ at muscle 6/7 in hemisegment A3. The null allele *Syx4^73^* exhibited a strong reduction in the number of synaptic boutons compared to control animals (*Figure 3A,B,G*). When *Syx4^73^* was placed in *trans* with a large chromosomal deficiency that removed the entire *Syx4* locus, a similar phenotype was observed compared to *Syx4^73^* mutants alone (*Figure 3G*), consistent with *Syx4^73^* being a null allele. *Syx4^48^* animals also exhibited a decrease in bouton number compared to controls (*Figure 3C,G*), which was less severe than the null mutant. The smallest deletion, *Syx4^39^*, had no change in bouton number compared to controls (*Figure 3D,G*). These results indicate that *Syx4* is required for normal synaptic bouton number. Furthermore, as the *Syx4^39^* allele lacks the start site for *Syx4A* and does not exhibit any phenotype, we hypothesize that expression of *Syx4B* from the second start site is sufficient for normal *Syx4* function with respect to bouton number.

We next overexpressed *UAS-Syx4A* or *UAS-Syx4B*, in the presynaptic neuron (with *elav-GAL4*) or the postsynaptic muscle cell (with *24B-GAL4*). None of these overexpressions resulted in any change in synaptic bouton number compared to control animals (*Figure 3G*). Thus, overexpression of *Syx4* is not detrimental to synaptic growth.

We next attempted to rescue *Syx4* mutant defects by expressing either isoform in the null mutant background with either pre- or postsynaptic-specific drivers at the NMJ. When expressed in the postsynaptic cell, both *Syx4A* and *Syx4B* were able to rescue the *Syx4^73^* decrease in bouton number compared to *Syx4^73^* alone (*Figure 3E–G*), restoring bouton number to control levels. In contrast, expressing either of these constructs presynaptically did not produce any rescue of the null mutant phenotype compared to *Syx4^73^* alone (*Figure 3G*). These findings indicate that *Syx4* is required postsynaptically to regulate bouton number. Furthermore, either isoform of *Syx4* is sufficient for *Syx4* function.

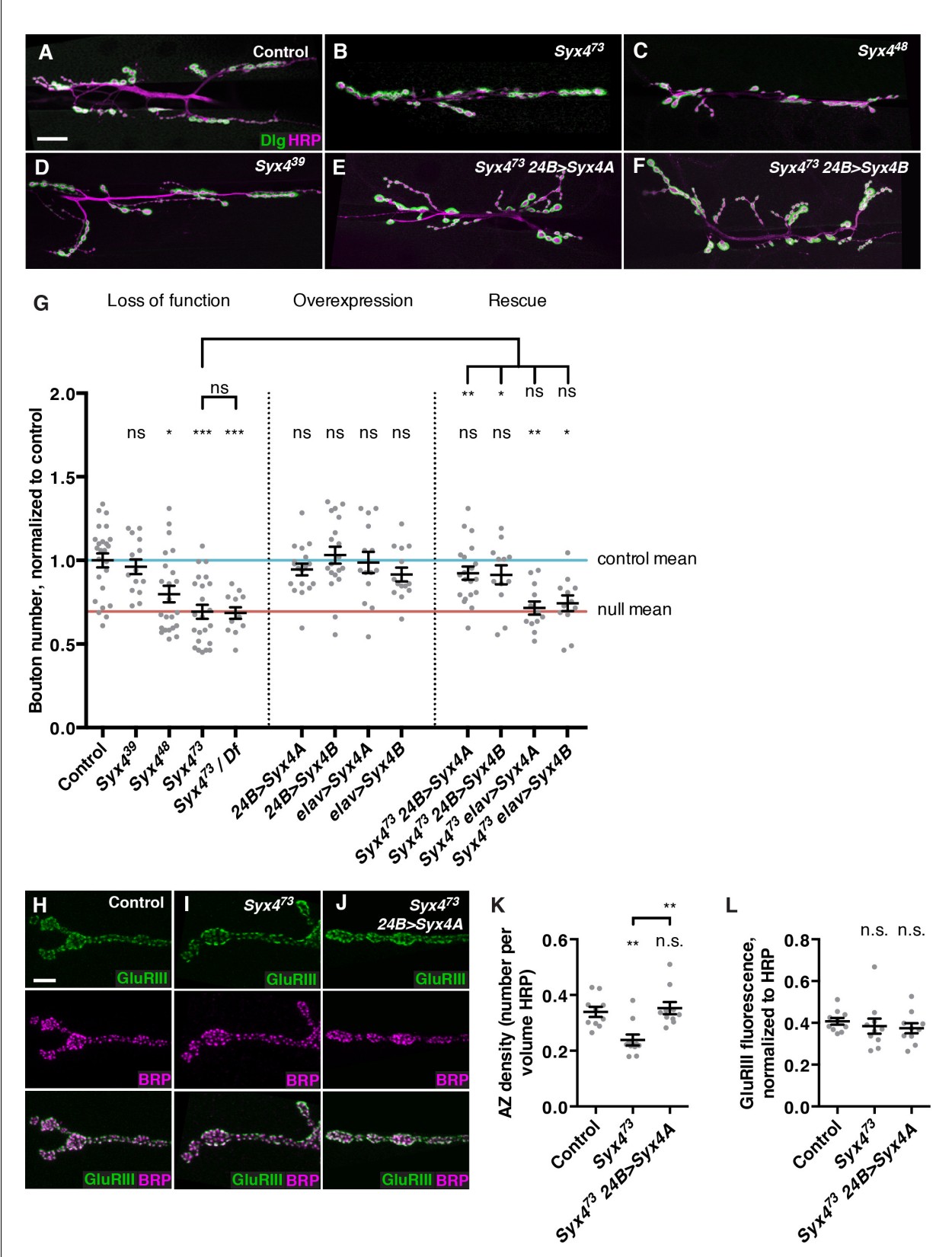

**Figure 3.** Syntaxin 4 regulates synaptic growth at the NMJ. (**A–F**) Representative images of NMJs stained with antibodies to the postsynaptic marker Dlg (green) and the neuronal membrane marker HRP (magenta) to highlight the number of synaptic boutons; images are shown from precise excision

*Figure 3 continued on next page*

*Figure 3 continued*

control (A), *Syx4^73* (B), *Syx4^48* (C), *Syx4^39* (D), *Syx4^73 24B>Syx4A* (E), and *Syx4^73 24B>Syx4B* (F) animals. (G) Quantification of bouton number, normalized to controls. Blue line indicates the control mean. Red line indicates *Syx4^73* null mean. Data are presented as mean ± SEM. (H–J), Representative images of NMJs stained with antibodies to GluRIII (green) and the AZ marker Brp (magenta); images are shown from precise excision control (H), *Syx4^73* (I), and *Syx4^73 24B>Syx4A* (J) animals. (K), Quantification of AZ density, calculated as the number of AZs per volume HRP. Data are presented as mean ± SEM. (L) Quantification of GluRIII fluorescence per HRP fluorescence. Data are presented as mean ± SEM. Scale bars = 20 μm (A–F), 5 μm (H–J). Statistical comparisons are fully described in Figure 3—source data 1, and are indicated here as ***p<0.001, **p<0.01, *p<0.05, ns = not significant; comparisons are with control unless indicated.

The following source data is available for figure 3:

**Source data 1.** Statistical data for *Figure 3*.

## Syx4 regulates active zone number

We also examined the organization of neurotransmitter release sites in *Syx4* mutants by staining for Bruchpilot (Brp), a marker of the presynaptic AZ, and GluRIII, an obligate subunit of the postsynaptic glutamate receptor. By counting Brp+ puncta, we detected a significant decrease in the density of AZs per unit volume in *Syx4* mutants compared to controls (*Figure 3H,I,K*). The AZ density defect was rescued by postsynaptic overexpression of Syx4 (*Figure 3J,K*). The amount of GluRIII fluorescence was unchanged in *Syx4* mutants compared to controls, indicating normal amounts of GluRIII were present at the postsynaptic membrane (*Figure 3H–J,L*). In addition, the apposition of Brp and GluRIII was unaffected (*Figure 3H–J*). Thus, *Syx4* mutants have a decrease in the density of AZs and a reduction in the total number of boutons, but no defects in the organization of individual release sites. The observation that postsynaptic Syx4 can regulate AZs in the presynaptic cell supports the hypothesis that Syx4 participates in retrograde signaling.

## Genetic interactions between Synaptotagmin 4 and Syntaxin 4

Syx4 regulates the membrane localization of Syt4, and loss of either gene leads to a reduction in the number of synaptic boutons at the larval NMJ (*Figure 3*; *Barber et al., 2009*). To further investigate the relationship between *Syt4* and *Syx4* in synaptic development, we tested for genetic interactions between the null allele *Syx4^73* and the null allele *Syt4^BA1* (*Adolfsen et al., 2004*). Single heterozygotes of *Syx4* (*Syx4^73/+*) or *Syt4* (*Syt4^BA1/+*) had no bouton number phenotype compared to control animals (*Figure 4A,B,F*). Strikingly, double heterozygotes (*Syx4^73/+ Syt4^BA1/+*) had a strong decrease in bouton number compared to control animals and compared to single *Syx4^73/+* or *Syt4^BA1/+* heterozygotes (*Figure 4C,F*). This finding is consistent with *Syx4* and *Syt4* acting together to regulate bouton number and synaptic growth.

To investigate the epistatic relationship between *Syx4* and *Syt4*, we produced animals that are hemizygous for *Syx4^73*, which is on the X chromosome, and homozygous for *Syt4^BA1* (*Syx4^73 Syt4^BA1*). These animals die during the 2^nd larva instar, precluding analysis at the 3^rd larval instar NMJ. As *Syx4^73* hemizygotes die during the pupal stage, and *Syt4^BA1* homozygotes survive to adulthood, the early lethality observed in the double mutants reveals a synergistic interaction between the genes. This indicates that *Syx4* and *Syt4* act in parallel pathways, rather than a single epistatic pathway.

We also produced animals that were heterozygous for one gene and homo/hemizygous for the other. All of these animals survived to the 3^rd larval instar, allowing us to assess bouton number. Removing one copy of *Syt4^BA1* in the *Syx4^73* hemizygous background did not modify the *Syx4^73* hemizygous phenotype (*Figure 4E,F*). In contrast, removing one copy of *Syx4^73* in the *Syt4^BA1* homozygous background resulted in a further reduction in bouton number compared to *Syt4^BA1* homozygotes alone (*Figure 4D,F*). Taken together with the observation that *Syx4* regulates membrane localization of Syt4, we conclude that *Syx4* and *Syt4* interact in one pathway, and also in parallel pathways, to regulate synapse development at the *Drosophila* NMJ.

## Syntaxin 4 interacts with Neurexin and Neuroligin

Based on the following observations, we hypothesize that Syx4 regulates the release of retrograde signals to control synaptic development: 1) Syx4 localizes to the postsynaptic membrane; 2) Syx4

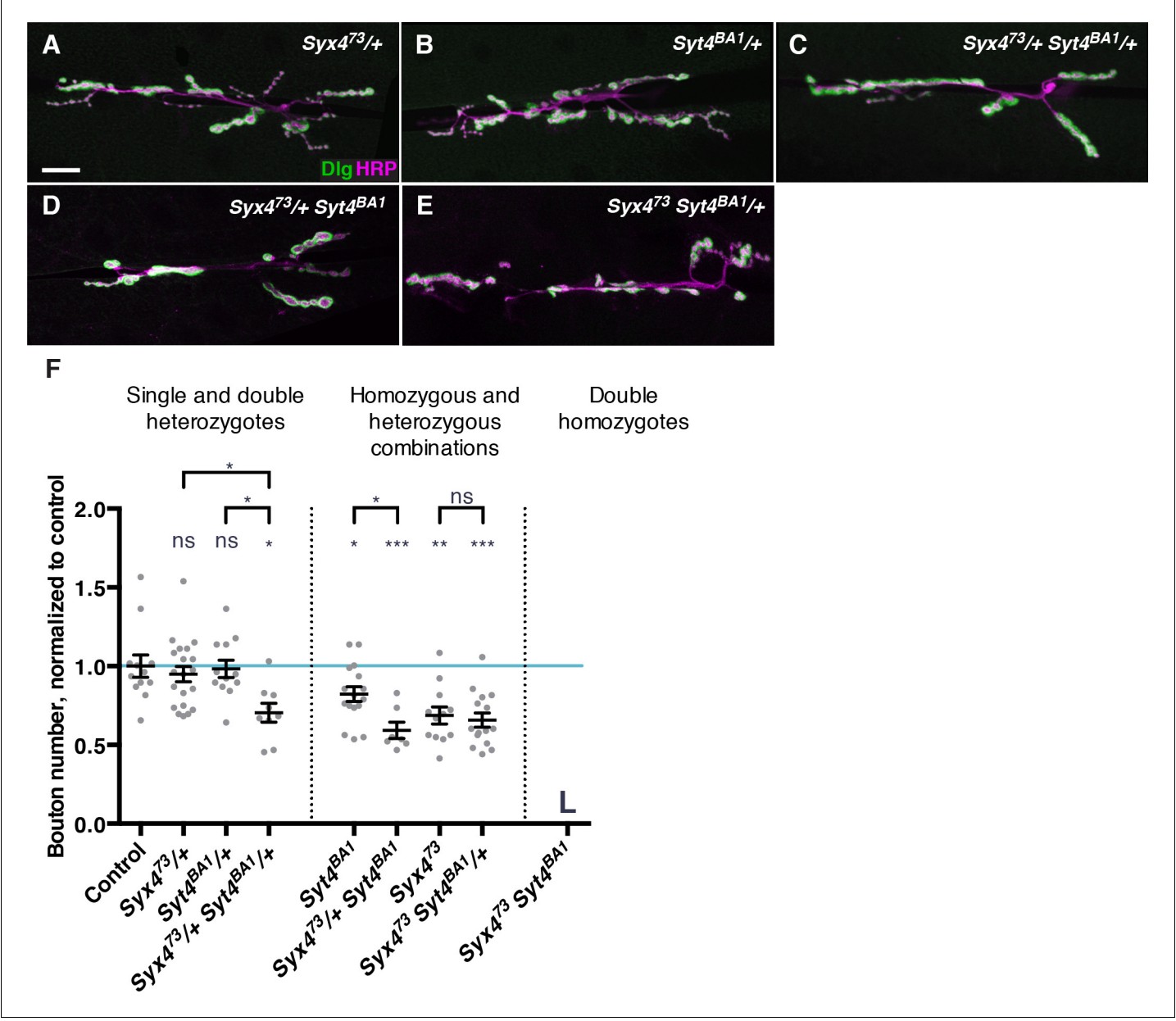

**Figure 4.** Genetic interactions between Syntaxin 4 and Synaptotagmin 4. (A–E) Representative images of NMJs stained with antibodies to the postsynaptic marker Dlg (green) and the neuronal membrane marker HRP (magenta) to highlight the number of synaptic boutons; images are shown from $Syx4^{73}/+$ (A), $Syt4^{BA1}/+$ (B), $Syx4^{73}/+$ $Syt4^{BA1}/+$ (C), $Syx4^{73}/+$ $Syt4^{BA1}$ (D), and $Syx4^{73}$ $Syt4^{BA1}/+$ (E) animals. (F) Quantification of bouton number, normalized to controls. Blue line indicates the control mean. Data are presented as mean ± SEM. L = lethal. Scale bars = 20 µm (A–E). Statistical comparisons are fully described in **Figure 4—source data 1**, and are indicated here as ***p<0.001, **p<0.01, *p<0.05, ns = not significant; comparisons are with control unless indicated.

The following source data is available for figure 4:

**Source data 1.** Statistical data for **Figure 4**.

regulates the membrane localization of Syt4; 3) Syx4 regulates bouton number both with and independently of Syt4; 4) postsynaptic Syx4 regulates presynaptic AZ density; and 5) Syx4 proteins have a conserved function as plasma membrane t-SNAREs. To identify retrograde signals potentially regulated by Syx4, we tested for genetic interactions between $Syx4^{73}$ and components of characterized retrograde signaling pathways that affect bouton number at the NMJ. We failed to detect any

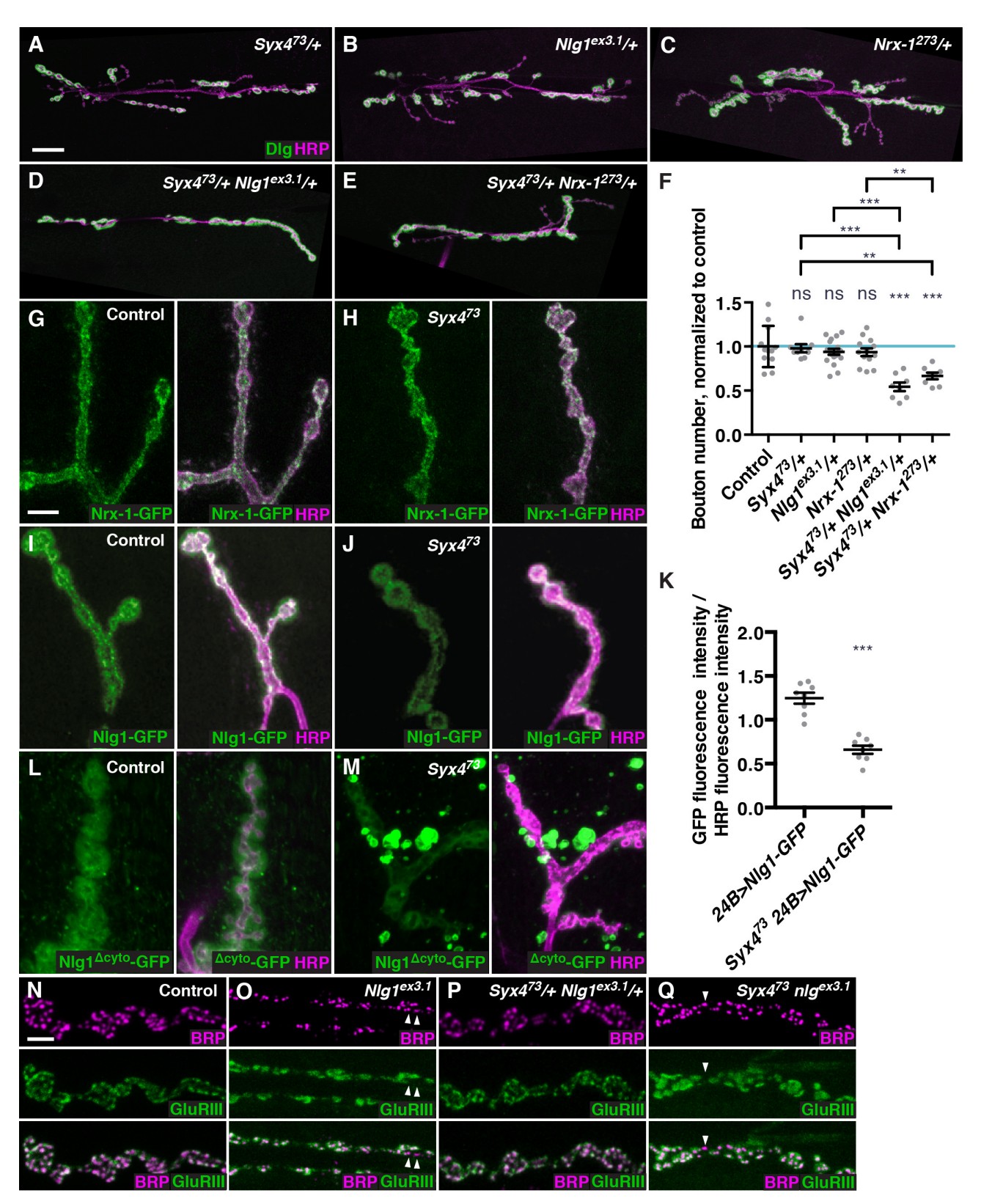

**Figure 5.** Syntaxin 4 interacts with Neuroligin 1 and regulates its membrane localization. (**A–E**), Representative images of NMJs stained with antibodies to the postsynaptic density marker Dlg (green) and the neuronal membrane marker HRP (magenta) to highlight the number of synaptic boutons; *Figure 5 continued on next page*

*Figure 5 continued*

images are shown from *Syx4^73^/+* (A), *Nlg1^ex3.1^/+* (B), *Nrx-1^273^/+* (C), *Syx4^73^/+ Nlg1^ex3.1^/+* (D), and *Syx4^73^/+ Nrx-1^273^/+* (E) animals. (F) Quantification of bouton number, normalized to controls. Blue line indicates the control mean. Data are presented as mean ± SEM. (G–H), Representative images of NMJs stained with antibodies against HRP (magenta) and expressing Nrx-1-GFP in a control (G) or *Syx4^73^* mutant (H) background. (I–J) Representative images of NMJs stained with antibodies against HRP (magenta) and expressing Nlg1-GFP in a control (I) or *Syx4^73^* mutant (J) background. (K) Quantification of GFP fluorescence per HRP fluorescence from animals expressing Nlg1-GFP in a control or *Syx4^73^* mutant background. Data are presented as mean ± SEM. (L–M) Representative images of NMJs stained with antibodies against HRP (magenta) and expressing Nlg1^Δcyto^-GFP in a control (L) or *Syx4^73^* mutant (M) background. (N–Q) Representative images of NMJs stained with antibodies against Brp (magenta) and GluRIII (green), from precise excision control (N), *Nlg1^ex3.1^* (O), *Syx4^73^/+ Nlg1^ex3.1^/+* (P), and *Syx4^73^ Nlg1^ex3.1^* (Q) animals. Arrowheads indicate AZs lacking an apposed GluR field. Scale bars = 20 μm (A–E), 5 μm (I,J,L–Q). Statistical comparisons are fully described in *Figure 5—source data 1*, and are indicated here as ***p<0.001, **p<0.01, *p<0.05, ns = not significant; comparisons are with control unless indicated.

The following source data and figure supplements are available for figure 5:

**Source data 1.** Statistical data for *Figure 5*.
**Figure supplement 1.** Genetic interaction experiments between *Syx4* and BMP pathway components.
**Figure supplement 2.** Genetic interaction experiments between single and double null mutants of *Syx4, Nlg1*, and *Nrx-1*.

dosage-dependent genetic interactions between *Syx4* and components of the retrograde BMP signaling pathway that affects arbor size, neurotransmitter release, and synaptic plasticity (*Figure 5—figure supplement 1*; *Aberle et al., 2002*; *Marqués et al., 2002*; *McCabe et al., 2003*; *Piccioli and Littleton, 2014*; *Rawson et al., 2003*).

In contrast, we detected strong genetic interactions between *Syx4* and the genes encoding the adhesion proteins Neurexin 1 (Nrx-1) and Neuroligin 1 (Nlg1) (*Figure 5*). Neurexins and Neuroligins form transsynaptic adhesion complexes, with a Neurexin typically the presynaptic partner and a Neuroligin the postsynaptic partner. At the *Drosophila* NMJ, Nrx-1 and the three characterized Nlgs (Nlg1-3) have been shown to play several roles in synaptic growth and organization, including regulation of bouton number, GluR subunit composition, and AZ size, spacing, and apposition (*Banovic et al., 2010*; *Chen et al., 2010, 2012*; *Li et al., 2007b*; *Sun et al., 2011*; *Xing et al., 2014*).

We used the null alleles *Nrx-1^273^* (*Li et al., 2007b*) and *Nlg1^ex3.1^* (*Banovic et al., 2010*) to test for interactions with *Syx4*. Single heterozygotes of *Nlg1* (*Nlg1^ex3.1^/+*), *Nrx-1* (*Nrx-1^273^/+*), and *Syx4* (*Syx4^73^/+*) all had a normal number of boutons compared to control animals (*Figure 5A–C,F*). However, the double heterozygotes *Syx4^73^/+ Nlg1^ex3.1^/+* (*Figure 5D*) and *Syx4^73^/+ Nrx-1^273^/+* (*Figure 5E*) exhibited strong reductions in bouton number compared to controls and compared to each single heterozygote (*Figure 5F*). Thus, Syx4, Nrx-1, and Nlg1 cooperate to regulate bouton number at the NMJ.

We next tested whether the localization of Nrx-1 or Nlg1 was perturbed upon loss of *Syx4*. We expressed GFP-tagged forms of Nrx-1 and Nlg1 (*Banovic et al., 2010*) at the synapse and measured fluorescence intensity in control and *Syx4* mutant backgrounds. When we expressed Nrx-1-GFP in the presynaptic cell using *elav-GAL4*, we did not detect any change in fluorescence intensity of GFP in *Syx4^73^* mutants compared to controls (*Figure 5G,H*). However, when we expressed Nlg1-GFP in the postsynaptic cell using *24B-GAL4*, we detected a significant reduction in GFP signal at the synapse in the *Syx4^73^* null mutant background compared to controls (*Figure 5I–K*). This result indicates that Syx4 regulates the levels of Nlg1 at the postsynaptic membrane.

If the amount of Nlg1 at the synapse depends on Syx4, it is possible that the cytoplasmic domain of Nlg1 is involved in its delivery or retention. To test this hypothesis, we examined the localization of a tagged Nlg1 construct lacking the cytoplasmic domain (Nlg1^Δcyto^-GFP; *Banovic et al., 2010*). This construct was previously shown to localize to the NMJ, and to produce a dominant negative decrease in bouton growth (*Owald et al., 2010*). Like the full-length construct, Nlg1^Δcyto^-GFP localizes to the postsynaptic membrane when expressed in a control background (*Figure 5L*). Interestingly, this localization pattern is strikingly different when Nlg1^Δcyto^-GFP is expressed in a *Syx4^73^* mutant background. While some tagged protein is observed at the synapse, Nlg1^Δcyto^-GFP also appears in prominent bright clusters seen both near the NMJ (*Figure 5M*) and throughout the

muscle (data not shown). This finding indicates a strong effect of Syx4 on Nlg1 localization, which is enhanced when the cytoplasmic domain of the protein is absent.

To further investigate the relationship between *Syx4*, *Nrx-1*, and *Nlg1*, we produced *Syx4$^{73}$ Nlg1$^{ex3.1}$* and *Syx4$^{73}$ Nrx-1$^{273}$* double mutant animals. Analysis of the double mutants revealed a strong reduction in bouton number compared to controls, and double mutants were not significantly different from single *Nlg-1* or *Nrx-1* mutants (**Figure 5—figure supplement 2**). Thus, complete loss of *Syx4* does not enhance the bouton formation defects seen in *Nlg1* or *Nrx-1* homozygous mutants. This observation suggests that *Nlg1* and *Nrx-1* are downstream of *Syx4* with respect to bouton number.

We next examined the organization of postsynaptic densities, which is perturbed in *Nlg1* mutants (**Banovic et al., 2010**), but not in *Syx4$^{73}$* (**Figure 3**). Consistent with previous studies (**Banovic et al., 2010**), we detected irregular and enlarged GluR fields in *Nlg1* mutants, as well as defects in apposition between AZs and GluR fields, compared to controls (**Figure 5N,O**, arrowheads). We then tested whether loss of *Syx4* could modify the *Nlg1* AZ/GluR phenotypes. We first tested animals that were double heterozygotes for *Syx4$^{73}$* and *Nlg1$^{ex3.1}$*, and observed normal GluR field size and apposition (**Figure 5P**). Thus, the dosage-dependent genetic interactions we detected with respect to bouton number are not seen in the case of AZ/GluR organization. Double mutant animals (*Syx4$^{73}$ Nlg1$^{ex3.1}$*) looked qualitatively similar to single *Nlg1$^{ex3.1}$* mutants (**Figure 5Q**, arrowheads), indicating that loss of *Syx4* did not modify this aspect of the *Nlg1* phenotype.

In summary, *Syx4* and *Nlg1* mutants have phenotypes that partially overlap (bouton number), but *Nlg1* mutants have additional defects not seen with loss of *Syx4* (organization of AZs/GluRs). Our genetic interaction data are consistent with *Syx4* and *Nlg1* cooperating to regulate bouton number, but not AZ/GluR organization. Taken together with the observation that loss of Syx4 leads to a partial reduction of Nlg1 at the membrane, one model is that minimal levels of Nlg1 are sufficient for AZ/GluR organization, but higher Syx4-dependent surface expression is required for regulating synaptic growth and bouton number.

## Neuroligin 1 mobility is not affected by loss of *Syntaxin 4*

How does loss of *Syx4* result in lower levels of Nlg1 at the postsynaptic membrane? One possibility is that less Nlg1 is delivered, and another is that Nlg1 is less stable or more mobile once it gets to the membrane. To investigate these possibilities, we measured the mobility of Nlg1 in vivo by tagging it with a photoconvertible fluorophore, Dendra2 (**Adam et al., 2009**; **Gurskaya et al., 2006**; **Figure 6A**). We added the Dendra2 tag in a juxta-membrane position, as previously described for Nlg1-GFP (**Banovic et al., 2010**). When expressed in the postsynaptic compartment using the muscle driver *24B-GAL4*, Nlg1-Dendra2 localized to the synapse similarly to Nlg1-GFP (**Figure 6B**). We then used a 405 nm laser to convert approximately 50% of the fluorescent signal in a single bouton (**Figure 6C,C'**), and followed the fluorescence intensity of the green (non-photoconverted; NPC) and red (photoconverted; PC) signals over a 10 min period (**Figure 6D** and **Video 1**). We measured mobility by calculating the relative change in fluorescence ($\Delta F/F$) in both channels from immediately after PC ($t_{1\ min}$) to 9 min after PC ($t_{10\ min}$), in the PC bouton (ROI1) or adjacent NPC boutons (ROI2 and ROI3), after correcting for photobleaching (**Figure 6C'–E**). If Nlg1-Dendra2 moved laterally in the membrane, we would expect to see a decrease in red/PC signal in ROI1 and an increase in red/PC signal in ROIs 2 and 3. If Nlg1-Dendra2 was internalized from the membrane, we would expect to see a decrease in red/PC signal in ROI1 without any increase in red/PC signal in ROIs 2 and 3. Interestingly, we measured extremely small $\Delta F/F$ values (< 0.02%) for the red/PC signal in all ROIs, reflecting very little change in fluorescence over the time course of the experiment (**Figure 6D,E**). We performed the same experiment by expressing Nlg1-Dendra2 in the *Syx4$^{73}$* mutant background and observed a similar effect, with small $\Delta F/F$ values in the red/PC channel, and no significant change compared to control values (**Figure 6E** and **Video 2**). Thus, over the time course measured, Nlg1 is immobile at the synapse, and this is not changed by loss of *Syx4*. We also monitored the fluorescence of green/NPC molecules, and measured very small $\Delta F/F$ values (< 0.02%) in all ROIs for both controls and *Syx4$^{73}$* mutants (**Figure 6D,E**). It is more difficult to interpret the movement of NPC molecules in this experiment; however, given the conclusion from the red/PC channel data that Nlg1 does not move laterally in the membrane or get internalized from the membrane over the time course of the experiment, the stable fluorescence of NPC molecules allows us to infer that very little new unconverted Nlg1 is delivered to the synapse. Thus, our analysis of photoconvertible Nlg1

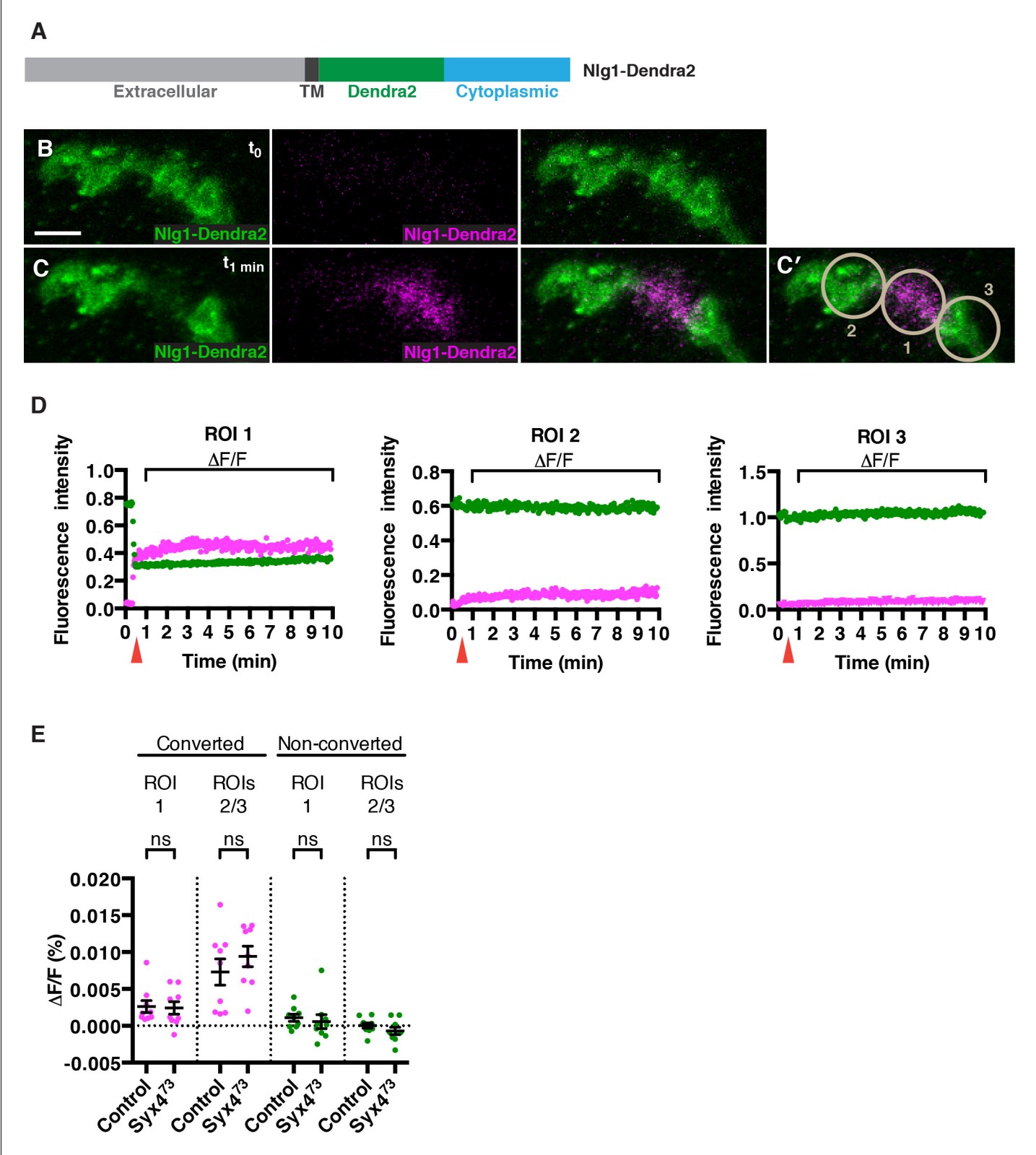

**Figure 6.** No change in mobility of Neuroligin 1 is observed in *Syntaxin 4* mutants. (**A**) Nlg1-Dendra2 construct. The Dendra2 tag was placed between the transmembrane domain and the cytoplasmic tail. (**B–C**) Representative image from a single animal expressing Nlg1-Dendra2 in the postsynaptic cell. One bouton (ROI1) was targeted with a 405 nm laser for photoconversion of the Dendra2 tag after 1 min. Non-photoconverted Nlg1-Dendra2 is shown in green and photoconverted Nlg1-Dendra2 is shown in magenta before (**B**) and immediately after (**C**) photoconversion. (**C'**) Regions of interest:
*Figure 6 continued on next page*

*Figure 6 continued*

ROI1, photoconverted region; ROIs 2 and 3, adjacent regions. (**D**) Fluorescent intensity over time for photoconverted and non-photoconverted molecules in all three ROIs. Red arrows indicate time of photoconversion. (**E**) Quantification of ΔF/F of both photoconverted and non-photoconverted molecules, in all three ROIs, in both the control and *Syx4*[73] mutant backgrounds. Data are presented as mean ± SEM. Scale bars = 5 μm. Statistical comparisons are fully described in **Figure 6—source data 1**; no significant differences found (ns = not significant).

The following source data is available for figure 6:

**Source data 1.** Statistical data for **Figure 6**.

reveals that Nlg1 is stable at the synapse over a short time course and this stability is not compromised by loss of *Syx4*. Based on these results, we hypothesize that the lower plasma membrane level of Nlg1 in *Syx4* mutants is the result of changes in the delivery or removal of Nlg1 over a developmental time scale, or over a longer time course than our experimental paradigm.

## Syntaxin 4, Synaptotagmin 4, and Neuroligin 1 regulate acute structural plasticity at the NMJ

In addition to synaptic growth during development, the *Drosophila* NMJ displays acute structural plasticity where new boutons bud rapidly in response to strong neuronal stimulation (*Ataman et al., 2008*). Newly formed boutons, called ghost boutons (GBs), are readily identifiable as round structures containing neuronal membrane, but without any postsynaptic apparatus. The activity-dependent budding of GBs requires retrograde BMP signaling (*Piccioli and Littleton, 2014*), as well as retrograde signaling mediated by Syt4 (*Korkut et al., 2013*; *Piccioli and Littleton, 2014*).

We investigated whether Syx4 regulates acute structural plasticity in vivo using a high K[+] stimulation protocol (*Ataman et al., 2008*; *Piccioli and Littleton, 2014*). As previously described, control animals exhibited robust GB budding in response to spaced incubations in high K[+] over a 30 min period (*Figure 7A,A′,K*). However, budding was strongly suppressed in *Syx4* null mutant animals compared to controls (*Figure 7E,E′,K*). Thus, like *Syt4*, *Syx4* regulates rapid activity-induced synaptic growth. Furthermore, *Syx4* and *Syt4* interact genetically with respect to GB budding, as budding was strongly suppressed in *Syx4*[73]/+ *Syt4*[BA1]/+ double heterozygotes compared to normal robust budding in either single heterozygote (*Figure 7B,B′,C,C′,H,H′,K*).

We next tested whether GB budding is impaired in *Nlg1* mutants, and whether *Syx4* and *Nlg1* interact in this context. Indeed, *Nlg1*[ex3.1] mutants exhibited a strong suppression of GB budding compared to controls (*Figure 7G,G′,K*). Also, GB budding was strongly suppressed in *Syx4*[73]/+

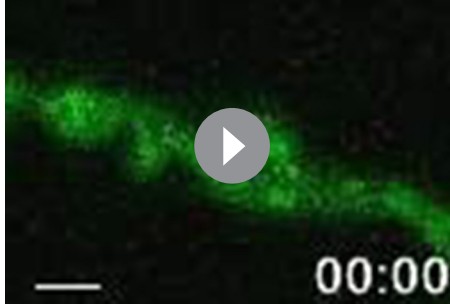

**Video 1.** Photoconversion of Nlg1-Dendra2 in control animals. Visualization of a synaptic arbor expressing postsynaptic Nlg1-Dendra2 at muscle 4 in a dissected third instar larva. One bouton is photoconverted after 20 sec, with about 50% of the green molecules converted to red (shown here as magenta). Over the next 9 min of imaging, very little movement of photoconverted molecules is observed. Scale bar = 2.5 μm.

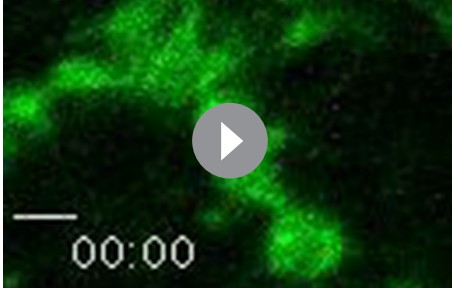

**Video 2.** Photoconversion of Nlg1-Dendra2 in *Syx4*[73] animals. Visualization of a synaptic arbor expressing postsynaptic Nlg1-Dendra2 at muscle 4 in a dissected third instar larva mutant for *Syx4*. One bouton is photoconverted after 20 sec, with about 50% of the green molecules converted to magenta. Over the next 9 min of imaging, very little movement of photoconverted molecules is observed. Scale bar = 2.5 μm.

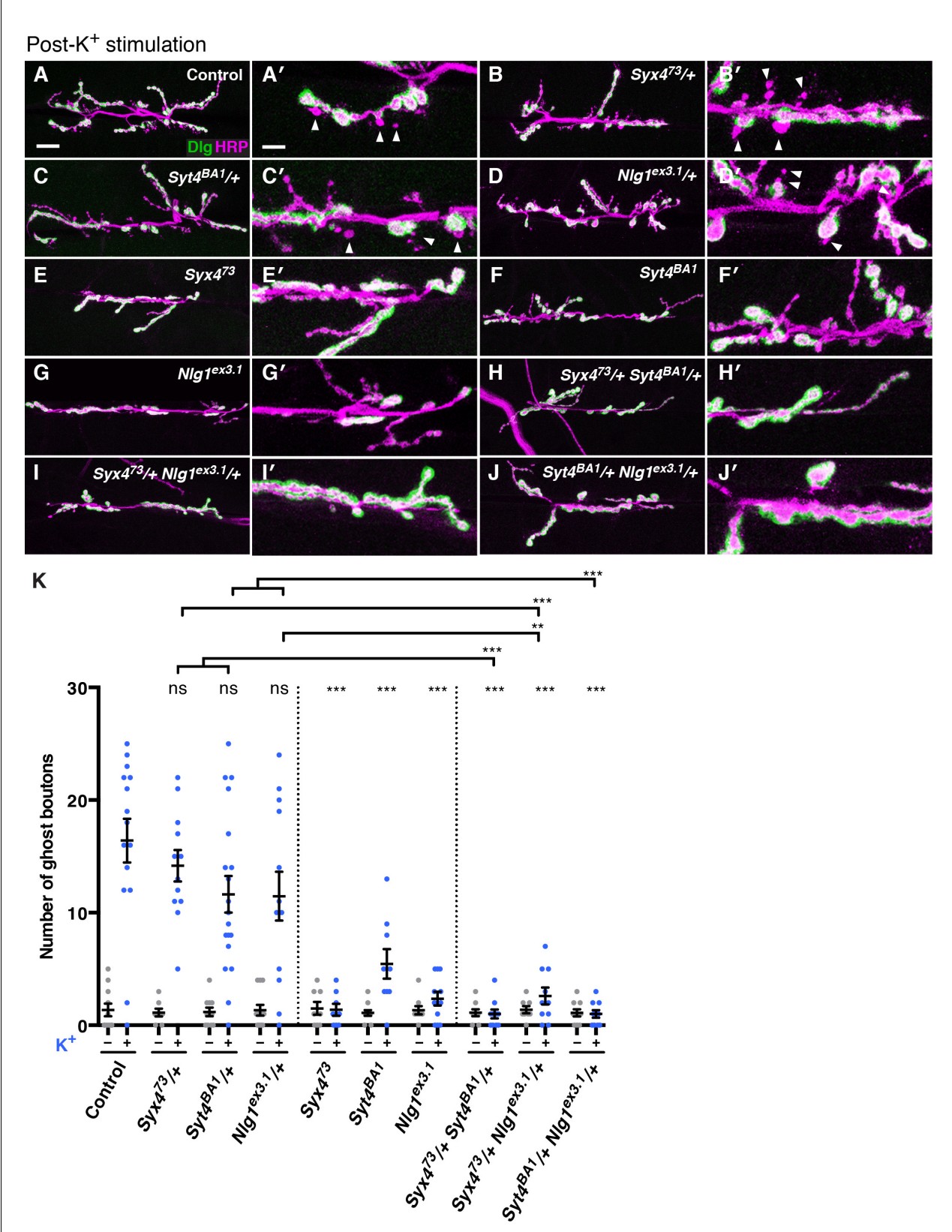

**Figure 7.** Syntaxin 4, Synaptotagmin 4, and Neuroligin 1 regulate acute structural plasticity at the NMJ. (A–J) Representative images of NMJs stained with antibodies to HRP (magenta) and the postsynaptic marker Dlg (green) to highlight synaptic boutons. Ghost bouton budding was stimulated with

*Figure 7 continued on next page*

*Figure 7 continued*
spaced incubations in high K$^+$. Ghost boutons are identified as round HRP+ structures lacking Dlg signal (arrowheads); images are shown from precise excision control (A), $Syx4^{73}$/+ (B), $Syt4^{BA1}$/+ (C), $Nlg1^{ex3.1}$/+ (D), $Syx4^{73}$ (E), $Syt4^{BA1}$ (F), $Nlg1^{ex3.1}$ (G), $Syx4^{73}$/+ $Syt4^{BA1}$/+ (H), $Syx4^{73}$/+ $Nlg1^{ex3.1}$/+ (I), and $Syt4^{BA1}$/+ $Nlg1^{ex3.1}$/+ (J) animals. (K) Quantification of ghost bouton number per NMJ from animals without (−) or with (+) high K$^+$ stimulation. Data are presented as mean ± SEM. Scale bars = 20 μm (A–J), 6.7 μm (A′–J′). Statistical comparisons are fully described in *Figure 7—source data 1*, and are indicated here as ***p<0.001, **p<0.01, *p<0.05, ns = not significant; comparisons are with control unless indicated.

The following source data and figure supplement are available for figure 7:

**Source data 1.** Statistical data for *Figure 5*.
**Figure supplement 1.** Interaction experiments between Syt4 and Nlg1.

$Nlg1^{ex3.1}$/+ double heterozygotes compared to single heterozygotes (*Figure 7C,C′,D,D′,I,I′,K*). These results indicate that Syx4 and Nlg1 interact to regulate activity-dependent formation of GBs.

Because Syx4 regulates the levels of both Syt4 and Nlg1 at the postsynaptic membrane, and all of these proteins are involved in regulating bouton number and rapid activity-dependent bouton formation, we investigated whether *Syt4* and *Nlg1* interact with each other. Indeed, we observed dosage-dependent genetic interactions between *Syt4* and *Nlg1* with respect to GB formation, as $Syt4^{BA1}$/+ $Nlg1^{ex3.1}$/+ double heterozygotes had a strong reduction in GB budding compared to single heterozygotes (*Figure 7C,C′,D,D′,J,J′,K*).

However, we were not able to detect an interaction between *Syt4* and *Nlg1* with respect to bouton number. The double heterozygotes $Syt4^{BA1}$/+ $Nlg1^{ex3.1}$/+ had a normal number of boutons compared to controls and compared to either single heterozygote (*Figure 7—figure supplement 1*), in contrast to the strong interactions we detected between *Syx4/Syt4* (*Figure 4*) and *Syx4/Nlg1* (*Figure 5*). We also expressed both Nlg1-GFP and Nlg1$^{Δcyto}$-GFP in the $Syt4^{BA1}$ null background and did not observe any change in Nlg1 localization compared to controls (*Figure 7—figure supplement 1*). Thus, our data are consistent with *Syx4, Syt4*, and *Nlg1* cooperating to regulate acute synaptic structural plasticity. With respect to bouton number, the data support *Syx4* interacting with *Nlg1* and *Syt4* in separate pathways. Taken together, Syx4 acts postsynaptically to regulate multiple parameters of synaptic biology by interacting with Nlg1 and Syt4 and regulating their membrane localization.

## Discussion

To identify regulators of postsynaptic exocytosis, we conducted a screen for gene products regulating Syt4 plasma membrane accumulation, resulting in the identification of the plasma membrane t-SNARE Syx4. Analysis of a *Syx4* null mutant indicates that Syx4 is essential for development of the *Drosophila* NMJ and regulates the membrane delivery of at least two proteins that are important for synaptic growth and plasticity: the postsynaptic Ca$^{2+}$ sensor Syt4 and the transsynaptic adhesion protein Nlg1.

### RNAi screen candidates suggest novel pathways that regulate retrograde signaling

Our screen identified 15 candidate gene products that altered the localization of Syt4-pH. In addition to Syx4, several other candidates motivate interesting hypotheses about regulatory pathways for postsynaptic exocytosis. MyoV is a Ca$^{2+}$-sensitive unconventional myosin that regulates polarized traffic (*Krauss et al., 2009*; *Li et al., 2007a*) and the release of exosomes from motorneurons (*Koles et al., 2012*). Thus, MyoV could play a role linking Ca$^{2+}$ influx to vesicle delivery or release at the synapse. Indeed, MyoV homologs have been implicated in regulated AMPA trafficking in mammalian dendrites (*Correia et al., 2008*; *Wang et al., 2008*). Two Rab regulators (Gdi and Rabex) suggest that key vesicle trafficking steps en route to the synapse are modulated by Rab activation states. Also, two cell adhesion molecules (Neuroglian and Contactin) indicate potential transsynaptic mechanisms regulating retrograde signaling. Neuroglian has been shown to be required for synaptic

stability (*Enneking et al., 2013*) and it is possible that Syt4-mediated retrograde signaling plays some role in this process.

Syt4 has also been shown to be transferred transsynaptically from the presynaptic terminal to the postsynaptic terminal on exosomes (*Korkut et al., 2013*). Thus, our approach of expressing Syt4-pH postsynaptically may not reveal components for the biosynthetic synthesis and transport of presynaptic Syt4. Nevertheless, the requirement for Syt4 in the postsynaptic cell for retrograde signaling is clear (*Barber et al., 2009*; *Korkut et al., 2013*; *Piccioli and Littleton, 2014*; *Yoshihara et al., 2005*), and the results of our screen highlight regulators of Syt4 trafficking to and from the postsynaptic membrane where Syt4 vesicles fuse in an activity-dependent manner (*Yoshihara et al., 2005*). The observation that endogenously expressed Syt4-GFP (Syt4$^{GFP-2M}$) shows a similar distribution to Syt4-pH supports the biological relevance of the screen data for identifying regulators of Syt4 trafficking in the postsynaptic cell.

## Syntaxin 4 regulates the localization of Syt4-pH and interacts with Syt4 to regulate bouton number

Our *Syx4* null allele phenocopies the *Syx4-RNAi* knockdown, reducing the delivery of Syt4-pH to the postsynaptic membrane. Consistent with this finding, loss of *Syx4* produces similar phenotypes to loss of *Syt4*. Both null mutants exhibit a reduction in the total number of boutons at the NMJ, indicating a defect in synaptic growth. Moreover, genetic interaction experiments clearly indicate that *Syx4* and *Syt4* interact with respect to synaptic growth. A strong genetic interaction between *Syx4* and *Syt4* is also evident at the level of lethality, as double mutant animals are lethal at a much earlier stage than either single mutant alone. Thus, even though Syx4 affects the localization of Syt4, suggesting they act in the same pathway, the genetic interaction data do not support a simple epistatic relationship. The difference in phenotypic severity, with the *Syx4* bouton number defect being significantly stronger than the *Syt4* defect, also points to Syt4 not being absolutely required for Syx4 signaling. A similar phenomenon is observed presynaptically where the t-SNARE Syx1 is indispensible for synaptic vesicle fusion, while fusion is only reduced in the absence of the synaptic vesicle Ca$^{2+}$ sensor Syt1. Taken together, we hypothesize that 1) Syx4 and Syt4 act together in a single pathway where Syx4 regulates the exocytosis of vesicles containing Syt4, and 2) Syx4 and Syt4 also act in divergent pathways, where Syt4 cooperates with other t-SNARES, and Syx4 mediates the exocytosis of vesicles in a Syt4-independent manner. This model allows for multiple possible postsynaptic SNARE complexes, regulating distinct release events. Dissecting the other components of these fusion machineries, and distinguishing activity-dependent from constitutive release events, will be important to build our understanding of how retrograde signaling is regulated.

## Syt4 regulates membrane levels of Neuroligin

In addition to affecting the localization of Syt4, *Syx4* mutants also exhibit a decrease in the amount of Nlg1 at the postsynaptic membrane. Nlg1 has several functions at the synapse, along with its presynaptic binding partner Nrx-1. Together they regulate bouton number as well as the size and spacing of active zones and glutamate receptors (*Banovic et al., 2010*; *Li et al., 2007b*; *Owald et al., 2010*), though some aspects of *Nlg1* signaling appear to be independent of *Nrx-1* (*Banovic et al., 2010*). Mutations in *Nrx* and *Nlg* family genes are also linked to ASD, highlighting the importance of Nrx-Nlg signaling in neuronal development (*Bottos et al., 2011*; *Südhof, 2008*). Consistent with a reduction of Nlg1 levels at the synapse, we observed strong genetic interactions between *Syx4*, *Nlg1* and *Nrx-1* with respect to bouton number. However, the prominent AZ/GluR defects seen in *Nlg1* and *Nrx-1* mutants were not observed in *Syx4* mutants, and heterozygous combinations did not produce these defects. It is likely that *Syx4* mutants exhibit a partial loss of function of *Nlg1*, and that bouton number is sensitive to this loss while AZ/GluR organization can be maintained with low levels of Nlg1.

A dramatic change in distribution of Nlg1$^{\Delta cyto}$ is observed in the *Syx4* mutant background, providing further evidence that Syx4 regulates the localization of Nlg1. The redistribution of Nlg1$^{\Delta cyto}$ to large accumulations is striking compared to full-length Nlg1, which is simply reduced at the synapse in the *Syx4* mutant background. This observation points to complex Syx4-dependent regulation of Nlg1 localization. One model is that trafficking of Nlg1 involves both a Syx4-dependent pathway and a second pathway that depends on an interaction with the Nlg1 C-terminus, which includes a

PDZ-domain-binding motif. In this scenario, a severe Nlg1 trafficking defect is revealed only when both pathways are compromised. A second possibility is that in the absence of Syx4, a portion of the Nlg1 content in the cell is degraded, but that this degradation step depends on the presence of the Nlg1 cytoplasmic tail, leading to the observed aggregation of Nlg1$^{\Delta cyto}$ in *Syx4* mutants.

Our analysis of Nlg1 trafficking in live animals reveals that Nlg1 is strikingly stable, in both control and *Syx4* mutant backgrounds. Our motivation in performing these experiments was to test possible mechanisms underlying the decrease in Nlg1 levels in *Syx4* mutants. It is possible that some Nlg1 mobility would be observed over a longer time course. Mammalian Nlg has been shown to undergo significant turnover at postsynaptic sites under LTP conditions in neuronal cell culture (*Schapitz et al., 2010*). Also, synaptic activity has been shown to induce cleavage of Nlg and the subsequent destabilization of the Nrx-Nlg complex (*Peixoto et al., 2012*). Thus, it remains a possibility that Nlg1 would be mobilized in response to activity in our preparation; however, we have not observed any increased mobility in response to high K$^+$ incubations in preliminary tests (data not shown). Our data are most consistent with Syx4 regulating Nlg1 over a developmental time course. A detailed examination of the relationship between Syx4 and Nlg1 dynamics will be crucial to understand how Syx4 contributes to this important pathway in synaptic development.

### Syx4, Syt4, and Nlg1 regulate synaptic plasticity

We observed a strong suppression of acute structural plasticity in null mutants of *Syx4, Syt4* and *Nlg1*. Double heterozygous combinations also indicated strong genetic interactions between all three of these genes with respect to plasticity. GB budding is regulated by both acute and developmental signaling. Because Syt4 postsynaptic vesicles fuse in an activity-dependent manner (*Yoshihara et al., 2005*), it is possible that Syt4-dependent signaling releases an acute instructive cue for GB budding. Thus, one attractive model is that Nlg1 is delivered to the membrane in response to stimulation, depending on the Ca$^{2+}$ sensitivity of Syt4 and the presence of the t-SNARE Syx4 at the membrane. It is also possible that Syx4-Syt4-Nlg1 signaling is required throughout development to potentiate the synapse to respond to strong neuronal stimulation. In conclusion, Syx4, Syt4, and Nlg1 interact to regulate several aspects of synaptic biology. Our data support multiple overlapping signaling pathways regulated by these proteins, reflecting a complex modulation of retrograde signaling to control synaptic growth and plasticity at the *Drosophila* NMJ.

## Materials and methods

### Drosophila stocks

All *Drosophila* strains were cultured on standard media at 25°C. The following stocks were used: *24B-GAL4* (BDSC 1767; *Brand and Perrimon, 1993*); *elav-GAL4[2]* (BDSC 8765; *Luo et al., 1994*); *Df(1)ED6630* (BDSC 8948; *Ryder et al., 2007*); *wit$^{A12}$, wit$^{B11}$* (*Marqués et al., 2002*); *UAS-Syt4-pHluorin* (*Yoshihara et al., 2005*); *Syt4$^{BA1}$* (*Adolfsen et al., 2004*); *gbb$^1$*(*Wharton et al., 1999*); *Nlg1$^{ex3.1}$, UAS-Nlg1-GFP, UAS-Nlg1$^{\Delta cyto}$-GFP* (*Banovic et al., 2010*); *Nrx-1$^{273}$* (*Li et al., 2007b*); *UAS-Syx4-RNAi* (TRiP JF01714; *Perkins et al., 2015*), *UAS-Syx4-RNAi* (VDRC 32413; *Dietzl et al., 2007*).

### Transgenics

Full-length Syntaxin 4 was obtained from the *Drosophila* Genomics Resource Center (DGRC RE02884; *Stapleton et al., 2002*). Three point mutations in the cDNA were corrected with a Quick-change Lightning Multi site-directed mutagenesis kit (Agilent Technologies, Santa Clara, CA) (pos 71: G to A; pos 496: A to C; pos 693: T to A). The sequence listed in Flybase, and several other ESTs covering parts of the Syx4 sequence, agree that these changes reflect the correct sequence. UAS-Syx4 was produced by PCR-amplifying Syx4 using ExTaq (ClonTech Laboratories, Mountain View, CA), and adding a 5'NdeI site and a 3'XbaI site. The PCR product was subsequently digested and subcloned into pValum (*Ni et al., 2008*). The construct was injected into a third chromosome docking strain (*y$^1$ w$^{67c23}$;P{CaryP}attP2*) by Best Gene Inc (Chino Hills, CA). UAS-RFP-Syx4 was produced by PCR-amplifying Syx4 and subcloning into pENTR/D-TOPO (Thermo Fisher Scientific, Waltham, MA). Syx4 was then moved into the destination vector pPRG using the Gateway system (Thermo Fisher Scientific; Gateway vectors developed by T. Murphy, The Carnegie Institution of

Washington, Baltimore, MD). The construct was injected into $w^{1118}$, along with a P-element helper plasmid, for random insertion by Best Gene Inc. Nlg1-Dendra2 was synthesized and subcloned into PBID-UASc (*Wang et al., 2012*) by Epoch Life Sciences (Sugar Land, TX). The Dendra2 tag (*Adam et al., 2009*; *Gurskaya et al., 2006*) was inserted between A865 and L866, about 11 aa downstream of the TM domain. These 11 aa were then repeated at the end of Dendra2, as previously described (*Banovic et al., 2010*). The construct was injected into a second chromosome docking strain ($y^1 w^{67c23}$; P{CaryP}attP40) by Best Gene Inc.

## Syx4 antibody production

Full length Syx4A was subcloned into pGEX-2T (GE Healthcare, UK) and GST-Syx4A protein was expressed and purified from OneShot BL21 cells (Thermo Fisher Scientific) as previously described (*Frangioni and Neel, 1993*). Rabbit immunosera were produced by SDIX (Newark, DE).

## Syx4 mutagenesis

The P element line P{EPgy2}Syx4[EY0005] (BDSC 14995; *Bellen et al., 2011*), carrying an insertion in the 5'-UTR of the Syx4 locus, was crossed to Tft/CyO, Δ2–3 (BDSC 8201) to mobilize the insertion. Single mosaic male progeny were then crossed to 2–4 females from the 1st chromosome balancer stock Df(1)ED6630/FM7i (BDSC 8948; *Ryder et al., 2007*). In the next generation, single white-eyed balanced females were crossed to 2–3 FM7i males. Approximately 150 lines were tested by PCR to detect deletions of the Syx4 locus. Three Syx4 alleles were identified and sequenced to determine the deletion breakpoints. $Syx4^{39}$: X:2743312..2743832 deleted and >1 kb of P-element sequence inserted; $Syx4^{48}$: X:2742469..2743832 deleted and ~690 bp of P-element sequence inserted; $Syx4^{73}$: X:2738999..[2750535–2752545] deleted. A precise excision was also identified and was used as a control line in all experiments unless otherwise indicated.

## Construction of the Syt4$^{GFP-2M}$ transgenic line

Homology-directed repair (HDR) following CRISPR/Cas9-induced double strand break was used to generate C-terminally tagged Syt4 knock-in lines. To construct the HDR donor plasmid pDsRed-Attp-syt4-DNA-eGFP, 1.1 kb of genomic DNA downstream of the Syt4 stop codon was inserted at the BglII site of pDsRed-Attp (Addgene #51019, gift from Melissa Harrison, Kate O'Connor-Giles, & Jill Wildonger), producing the plasmid pDsRed-Attp-Syt4-p2. Then 1.3 kb of genomic DNA upstream of the Syt4 stop codon was fused with eGFP coding sequence and inserted at the NheI site of pDsRed-Attp-syt4-p2, producing pDsRed-Attp-Syt4-DNA-eGFP-pre. Finally, the gRNA binding sites in this plasmid were mutated, resulting the final donor plasmid. All cloning steps were performed using Gibson Assembly (New England Biolabs, Ipswich, MA, #E5510).

Two gRNA sequences were designed according to *Gokcezade et al., 2014* and inserted into pCFD4-U6:1_U6:3tandemgRNAs (Addgene #49411; *Port et al., 2014*). The plasmid was injected into y1 w67c23; P{CaryP}attP40 embryos by Best Gene Inc. to generate the Syt4-gRNA stock.

To generate the GFP-tagged Syt4 flies, yw; nos-Cas9 flies (*Kondo and Ueda, 2013*) were crossed with Syt4-gRNA flies, and the embryos from the cross were injected with the donor plasmid. Successful transformants were screened for the presence of 3XP3-DsRed in the flies. The nos-Cas9 and Syt4-gRNA expression cassettes were crossed out in the next generation. In the final stock, PCR and sequencing were performed to confirm the insertion and verify that no mutation was present. Several independent lines were generated and validated. One of the homozygous viable and fertile lines, Syt4$^{GFP-2M}$, was used for all experiments. Homozygous animals were stained with antibodies against GFP to visualize Syt4$^{GFP-2M}$ protein.

## Immunostaining

Larvae were reared at 25°C and dissected at the third wandering instar stage. Larvae were dissected in HL3.1 solution (in mM, 70 NaCl, 5 KCl, 10 NaHCO$_3$, 4 MgCl$_2$, 5 trehalose, 115 sucrose, 5 HEPES, pH 7.2) and fixed in 4% paraformaldehyde or as otherwise indicated. Following washes in PBT (PBS containing 0.3% Triton X-100), larvae were blocked for one hour in PBT containing 2% normal goal serum, incubated overnight with primary antibody at 4°C, washed, incubated with secondary antibodies for 2 hr at room temperature, washed, and mounted in Vectashield (Vector Laboratories, Burlingame, CA) for imaging. For Syx4 stainings, Syx4 antibody was preabsorbed on

*Syx4* null mutant tissue to reduce background staining. Antibodies were as follows: mouse anti-Dlg, 1:1000 (DSHB 4F3; *Parnas et al., 2001*); anti-Brp, 1:500 (DSHB nc82; *Wagh et al., 2006*); anti-GluR-III, 1:500 (*Marrus et al., 2004*); anti-GluRIII-488, 1:500 (*Blunk et al., 2014*; *Marrus et al., 2004*); anti-Syx4, 1:500; rabbit anti-Lva, 1:500 (*Sisson et al., 2000*); DyLight 649 conjugated anti-horserad-ish peroxidase, 1:1000 (Jackson ImmunoResearch, West Grove, PA); Alexa Fluor 488 goat anti-mouse, Alexa Fluor 488 goat anti-rabbit, and Alexa Fluor 546 goat anti-mouse, 1:400 (Thermo Fisher Scientific). Images were acquired with a 40 × 1.3 NA oil-immersion objective (Carl Zeiss, Germany).

## RNAi screen

The recombinant stock *UAS-Syt4-pHluorin, 24B-GAL4* was produced and used for the screen. Females from this line were crossed to *UAS-RNAi* males, and 3 progeny were dissected per RNAi line tested. Larvae were dissected in HL3.1 buffer, fixed in 4% paraformaldehyde, washed in PBT, incubated overnight at 4°C with antibodies against HRP, washed in PBT, and mounted in Vectashield (Vector Laboratories). Syt4-pH distribution at the NMJ was analyzed in hemisegment A3 at muscle 4. A control cross was included in every batch, where *UAS-Syt4-pHluorin, 24B-GAL4* was crossed to a UAS line that had no effect on Syt4-pH distribution (*UAS-FLP*; BDSC 4540; *Duffy et al., 1998*). A list of all RNAi stocks screened is found in *Supplementary file 1*.

## High K$^+$ stimulation of larval NMJs

The acute structural plasticity assay was performed as previously described (*Piccioli and Littleton, 2014*). Wandering third instar larvae were dissected in HL3 solution (in mM, 70 NaCl, 5 KCl, 0.2 CaCl$_2$, 20 MgCl$_2$, 10 NaHCO$_3$, 5 trehalose, 115 sucrose, and 5 HEPES, pH7.2). Dissecting pins were then moved inward to 60% of the original size for each larva. Relaxed fillets were subjected to three 2 min incubations in high K$^+$ solution (in mM, 40 NaCl, 90 KCl, 1.5 CaCl$_2$, 20 MgCl$_2$, 10 NaHCO$_3$, 5 trehalose, 5 sucrose, and 5 HEPES, pH 7.2), spaced by 10 min in HL3. After the third high K$^+$ incuba-tion, larvae were returned to HL3 solution for 2 min and then stretched to their original size and fixed.

## Quantification of confocal images

Analyses were conducted using Volocity (version 6.3) or FIJI / ImageJ (version 2.0.0-rc-32/1.49v; *Schindelin et al., 2012*). Ghost boutons were identified by the presence of a presynaptic bouton (HRP–labeled) that lacked Dlg staining in fixed preparations. Counting of boutons and GBs was con-ducted at hemisegment A3 at muscle 6/7, and n refers to the number of NMJs analyzed, with no more than two NMJs analyzed per animal, and with animals derived from at least three independent experiments. Measurements of AZ density and GluR intensity were conducted on 12 1b boutons per animal, using 1 terminal bouton and 5 adjacent non-terminal boutons, on two different branches; n refers to the number of animals analyzed. AZ density was quantified manually by counting Brp–labeled puncta and dividing by the volume of HRP. GluR intensity was quantified by measuring the fluorescence intensity of GluRIII signal within an ROI defined by the HRP signal, and the average intensity within the ROI was divided by the average HRP intensity. All analyses were performed blind to genotype.

## Photoconversion of Nlg1-Dendra2

Wandering third instar larvae expressing postsynaptic Nlg1-Dendra2 were dissected in HL3.1 saline at room temperature. Images were acquired with a Carl Zeiss LSM 700 with a 40× 0.8 NA water-immersion objective using Zen software (Zeiss). A single confocal plane of a muscle 4 NMJ in hemi-segment A3 was continuously imaged for 10 min in the red and green channels. After the first 20 s, a single bouton was targeted with pulses of the 405 nm laser at 100% power until ∼50% of the green signal was converted. Images were stabilized using the Image stabilizer plug-in from FIJI / Image J (K. Li, The image stabilizer plugin for ImageJ, http://www.cs.cmu.edu/∼kangli/code/Image_Stabilizer.html, February, 2008). Fluorescence intensity was measured in the photoconverted ROI, two adjacent ROIs, and a 4$^{th}$ distant ROI (photobleaching ROI). Each intensity measurement was divided by the photobleaching ROI measurement at that time point to correct for photobleaching. $\Delta F/F$ was calculated on corrected measurements as $(F_{T10min}-F_{T1min})/F_{T1min}*100$.

## Statistical analyses

Statistical analyses were conducted using GraphPad Prism. Statistical significance in two-way comparisons was determined by a Student's *t*-test, while ANOVA analysis was used when comparing more than two datasets. The P values associated with ANOVA tests were adjusted P values obtained from a Tukey's post hoc test. In all figures, the data is presented as mean $\pm$ SEM; *** $p<0.001$, ** $p<0.01$, * $p<0.05$, n.s. not significant. Statistical comparisons are with control unless noted. Sample size (n), mean, SEM, and pairwise statistical comparisons are presented in figure supplements.

## RT-PCR analysis of Syntaxin 4

RNA was extracted from 5 larvae per sample using an RNease Mini kit (Qiagen Sciences, Germantown, MD) and treated with DNAse I (Qiagen). RT-PCR was carried out using a SuperScript One-Step RT-PCR System with Platinum Taq (Thermo Fisher Scientific). Forward primers were designed to bind to unique *5'-UTR* sequences of the *Syx4A* and *Syx4B* transcripts.

## Acknowledgements

This work was supported by NIH grant NS40296. We thank S Sigrist, H Aberle, M Bhat, O Papoulas, and A Vasin for sharing reagents. We thank the Bloomington *Drosophila* Stock Center (Indiana University, Bloomington, IN; NIH P40OD018537), the *Drosophila* Genome Resource Center (Indiana University, Bloomington, IN; NIH 2P40OD010949-10A1), the TRiP at Harvard Medical School (Boston, MA; NIH/NIGMS R01-GM084947), the Developmental Studies Hybridoma Bank (University of Iowa, Iowa City, IA), and the Vienna *Drosophila* Resource Center (Austria), for providing materials used in this study.

## Additional information

### Funding

| Funder | Grant reference number | Author |
| --- | --- | --- |
| National Institutes of Health | NS40296 | J Troy Littleton |

The funders had no role in study design, data collection and interpretation, or the decision to submit the work for publication.

### Author contributions

KPH, Conception and design, Acquisition of data, Analysis and interpretation of data, Drafting or revising the article; YVZ, ZDP, Acquisition of data, Analysis and interpretation of data; NP, Conception and design, Contributed unpublished essential data or reagents; JTL, Conception and design, Analysis and interpretation of data, Drafting or revising the article

### Author ORCIDs

Kathryn P Harris, http://orcid.org/0000-0003-1769-2587
J Troy Littleton, http://orcid.org/0000-0001-5576-2887

## Additional files

### Supplementary files

• Supplementary file 1. Candidate list for RNAi screen Complete list of transgenic lines tested. Each carries a UAS-RNAi to knock down the gene product (indicated with CG and gene symbol). RNAi stock ID refers to the unique ID from either the TRiP center at Harvard Medical School or the Vienna *Drosophila* Resource Center.

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
