## [Decision Letter]

Thank you for submitting your work entitled "The postsynaptic t-SNARE Syntaxin 4 controls traffic of Neuroligin and Synaptotagmin 4 to regulate retrograde signaling" for consideration by *eLife*. Your article has been favorably evaluated by Eve Marder (Senior editor) and three reviewers, one of whom, Hugo Bellen, is a member of our Board of Reviewing Editors.

The reviewers have discussed the reviews with one another and the Reviewing Editor has drafted this decision to help you prepare a revised submission.

Summary:

Harris et al. build on earlier work, showing that the *Drosophila* synaptotagmin Syt4 is necessary for a form of calcium-dependent homeostatic retrograde signalling at *Drosophila* synapses. They aim to dissect the postsynaptic signalling mechanisms, through a small-scale forward genetic RNAi screen for genes that affect postsynaptic Syt4 levels or localization.

From a screen of a few hundred genes encoding synaptic proteins, they identify 15 candidate genes, including the syntaxin Syx4 – knockdown of this causes apparent trafficking defects of Syt4, and reduces Syt4 levels at the postsynaptic membrane. Syx4 is enriched at the postsynaptic membrane, and Syx4 mutations generated by the authors show a similar phenotype on Syt4 localization as Syx4 knockdown, as well as reduced bouton numbers comparable to *Syt4* mutants, and a reduction in active zone density. The genetics is robust, with key mutant phenotypes shown using heteroallelic genotypes, knockdowns, and transgenic rescue obtained using muscle-specific expression.

Genetic interactions suggest that Syx4 and Syt4 affect the same process or pathway: *syx4/+ syt4/+* double heterozygotes have a more extreme phenotype than either single heterozygote, suggesting that they act together; and double homozygotes show much earlier lethality than single homozygotes, suggesting also action in parallel pathways; and 50% reduction of Syx4 in a Syt4 homozygous background caused a more severe phenotype than Syt4 homozygotes alone.

Similar genetic interactions with the postsynaptic neuroligin Nlg1 and its presynaptic binding partner Nrx1 suggest this as a route for retrograde Syx4-dependent signaling. Consistent with this model, *Syx4* mutants show lowered levels of postsynaptic Nlg1, but not presynaptic Nrx1. Photoconversion of an Nlg1::Dendra fusion shows that these lowered levels are not associated with altered levels of Nlg1 mobility. Finally, the authors show that a measure of structural synaptic plasticity, appearance of new boutons after high K^+^ stimulation, is abolished by loss of any of Syt4, Syx4 or Nlg1, and that double heterozygous mutant combinations show strong genetic interactions, suggesting that they affect a common pathway.

Taken together, the authors’ data confer a plausible mechanistic basis for the role of Syt4 in retrograde signalling, involving control of Syt4 localization by Syx4, and retrograde signalling via an Nlg1-Nrx1 interaction. This will be of wide interest. The strengths of the work are the robust genetics, the genetic interactions that show in vivo functional interactions between the players involved, the phenotypic analysis (with a caveat on the statistical analyses, see below), and the observed effects of Syx4 on localization of both Syt4 and Nlg1. Identification of additional players, and the mechanistic evidence for a plausible model, make the paper a significant advance, although one limitation is that they do not yet have direct molecular mechanisms for the phenotypes and functional interactions that they see.

Essential revisions:

The tagged Syt4-pH is an interesting reagent but the authors do not show that it rescues a Syt4 mutation and hence that is functional. It may also not reflect the endogenous localization of Syt4.

First, they should show that this construct rescues a *Syt4* mutant or not.

Second, the authors should show that the overexpressed Syt4 protein localization reflects the endogenous localization and that the various mutants also affect endogenous protein localization. I assume they did not raise an ab and there is probably not one available. Given that a significant portion of the manuscript is based on the Syt4 mislocalization, we suggest that the authors order the Syt4-GFP tagged fosmid made in Dresden by Sarov and Schnorrer (there is also a Syx4-GFP construct available) to assess if it exhibits the same protein localization as their construct (see http://biorxiv.org/content/early/2015/10/04/028308). Given that the gene is a genomically tagged with GFP its expression should reflect the entire expression of the gene in all tissues, not just localization in muscles. As it will not be overexpressed it should also be a useful reagent for other experiments as well.

Third, the *Syx4* mutants have an active zone number phenotype. The authors should show that Syx4 expression in muscles rescues this phenotype. It is a pre-synaptic phenotype that would really build the case for retrograde signaling.

The Discussion mentions a number of genes that were isolated in the screen. These data should be in the Results section as a prelude to the selection of Syx4, not in the Discussion. This may provide a better rationale for picking Syx4, which is currently non-existing.

When the authors observe synergystic phenotypic genetic interactions they should sometimes assess protein localizations of the target of interest as well. That will allow them to correlate the phenotype with the protein of interest. We are not requesting that they do this systematically but in a few key cases. Also, how come the number of ghost boutons is so different between this study and previous studies by the same group?

Can the authors visualize the release of vesicles post-synaptically using live imaging during ghost bouton formation?

Analyses requiring any manual or subjective steps (e.g. bouton number, defining ROI) must be performed blind to genotype. I could not find this information in the paper. If this has not been performed blind to genotype, the data should be reanalysed.

Analyses must be based on an adequate number of independent experiments. I could not find this information in the Materials and methods. For example, the bouton analysis is often stated n=8, n=12, etc. For independent experiments this is a reasonable number. However, if all 8 or 12 larvae came from the same vial (or even couple of vials), this is not acceptable – then there is no way to distinguish an effect of vial from an effect of genotype.

In the subsection “Nlg mobility is not affected by loss of *Syntaxin 4*” and Figure 6. The authors use M-W (U?) tests with n=4. This is a non-parametric test that considers only rank order and throws away quantitative data, making it not very effective at detecting significant effects. I don't see how n of 4 can be high enough to give such significant P values, and conversely, a non-significant P value can result from the low statistical power (and n) of this test. Even for a parametric test, an n of 4 is barely acceptable. The authors need a better way to interpret their data and draw robust conclusions, e.g. increase n, and/or compare a parameter that is normally distributed.

---

## [Author Response]

Essential revisions: The tagged Syt4-pH is an interesting reagent but the authors do not show that it rescues a Syt4 mutation and hence that is functional. It may also not reflect the endogenous localization of Syt4.

These are excellent suggestions to support the biological relevance of our screen. We have performed two major experiments to address these points. The first was to investigate whether Syt4-pH was functional by testing its ability to rescue *Syt4* mutants. Two prominent synaptic phenotypes that have been described in *Syt4* null animals are: 1) a decrease in the number of boutons at the NMJ, and 2) a reduction in the ability to bud new boutons (“ghost boutons”) in response to strong neuronal stimulation. We found that when Syt4-pH was expressed postsynaptically (with a muscle GAL4 driver) in the *Syt4* mutant background, the animals resembled control animals with respect to both of these parameters. Thus, postsynaptic Syt4-pH is functional at the NMJ and able to rescue these two well-described *Syt4* mutant phenotypes. These findings are shown in Figure 1—figure supplement 1.

Secondly, we created a C-terminal tagged Syt4 knock-in line (Syt4^GFP-2M^) using CRISPR/Cas9 technology to assess the endogenous localization of Syt4. This allows for normal regulatory control of Syt4 transcription (which has previously been demonstrated to be activity-regulated) and avoids potential overexpression artifacts. We found that the CRISPR tagged-Syt4 knock-in protein accumulates at the NMJ similarly to Syt4-pH. Endogenous Syt4^GFP-2M^ is also lost from the membrane in the *Syx4* mutant background, as seen with Syt4-pH. Thus, Syt4-pH localization recapitulates endogenous Syt4 distribution. These findings are shown in Figure 2. We also validated the Syt4^GFP-2M^ line in Figure 1—figure supplement 1, demonstrating that it is functional and does not exhibit any synaptic phenotypes related to loss of Syt4.

First, they should show that this construct rescues a Syt4 mutant or not. Second, the authors should show that the overexpressed Syt4 protein localization reflects the endogenous localization and that the various mutants also affect endogenous protein localization. I assume they did not raise an ab and there is probably not one available. Given that a significant portion of the manuscript is based on the Syt4 mislocalization, we suggest that the authors order the Syt4-GFP tagged fosmid made in Dresden by Sarov and Schnorrer (there is also a Syx4-GFP construct available) to assess if it exhibits the same protein localization as their construct (see http://biorxiv.org/content/early/2015/10/04/028308). Given that the gene is a genomically tagged with GFP its expression should reflect the entire expression of the gene in all tissues, not just localization in muscles. As it will not be overexpressed it should also be a useful reagent for other experiments as well. Third, the Syx4 mutants have an active zone number phenotype. The authors should show that Syx4 expression in muscles rescues this phenotype. It is a pre-synaptic phenotype that would really build the case for retrograde signaling.

We have performed this rescue experiment, and found that postsynaptic expression of Syx4 rescues the decrease in AZ density observed in *Syx4* mutants. This data was added to Figure 3).

*The Discussion mentions a number of genes that were isolated in the screen. These data should be in the Results section as a prelude to the selection of Syx4, not in the Discussion. This may provide a better rationale for picking Syx4, which is currently non-existing.*

Table 1 displays all 15 candidates identified in the screen. We also added some text to better describe our rationale for selecting Syx4 for further study, based on its conserved function as a plasma membrane t-SNARE, and its effect of severely decreasing Syt4-pH accumulation at the plasma membrane.

When the authors observe synergystic phenotypic genetic interactions they should sometimes assess protein localizations of the target of interest as well. That will allow them to correlate the phenotype with the protein of interest. We are not requesting that they do this systematically but in a few key cases.

As the reviewer points out, a major approach used in this study is the analysis of synergistic phenotypes to identify genetic interactions. Almost exclusively, these experiments involve creating double heterozygous animals and comparing them to single heterozygotes. For example, single heterozygotes of *Syx4 (Syx4/+*) and Syt4 (*Syt4/+*) each have a similar number of boutons as control animals, but double heterozygotes (*Syx/+ Syt4/+*) have a severe decrease in bouton number, suggesting a strong interaction between the genes (Figure 4). However, we don't think that assessing protein localizations in these animals would alter any interpretations of our data. For example, in the *Syx4/Syt4* double heterozygotes described above, we are not proposing that any specific target protein is mislocalized, but are rather using the phenotype as evidence that the genes interact in one or parallel pathways. Similarly, the strong phenotypes observed in the *Syx4/Nlg* double heterozygotes (Figure 5) suggested that these genes interact, which we then followed up by examining the protein localization of Nlg in the *Syx4* background (Figure 5). The read-out from every single genetic interactions experiment we performed is a complex phenotype regulated by multiple signaling pathways in the cell – the aim is to show that the phenotype can be modified by subtle combinatorial knock-down of two genes of interest, which then motivates follow-up to address specific mechanisms.

*Also, how come the number of ghost boutons is so different between this study and previous studies by the same group?*

A previous publication from our lab (Piccioli et al., 2014) reported an average of 6.6 ± 7.0 ghost boutons per NMJ in control animals, while in this study we report an average of 16.4 ± 1.9 ghost boutons per NMJ. The primary reason for this difference lies in the specific NMJs used for analysis: in the first study, NMJs from abdominal hemisegments A2 through A5 were included in analysis, while in the current study, we only analyzed NMJs from abdominal hemisegment A3. Both NMJ size and ghost bouton budding vary substantially between hemisegments, with hemisegment A2 typically having more boutons and more ghost boutons, and hemisegments A4 and A5 having fewer boutons and fewer ghost boutons, compared to A3. Both analyses are valid, as long as equal numbers of each hemisegment are included across genotypes in the first approach. For this study, our analysis of only A3 hemisegments yields a higher average of ghost bouton budding over all, and a smaller SEM value, and is the preferred approach for us going forward as we reduce variability associated with different size NMJs at distinct abdominal segments.

Can the authors visualize the release of vesicles post-synaptically using live imaging during ghost bouton formation?

We thank the reviewers for suggesting this experiment, which could provide exciting evidence of an acute link between postsynaptic exocytosis and ghost bouton formation. While Syt4-pH is a useful tool to visualize Syt4-positive domains at the postsynaptic membrane, we have not yet been successful in identifying fusion events during live imaging sessions. Our recently created Syt4-GFP knock-in line, *Syt4^GFP-2M^*, may also be useful to test this idea in the future.

We have looked extensively in fixed tissue, in order to examine the distribution of Syt4-pH relative to ghost bouton structures (see Figure 8). As previously described, ghost boutons are devoid of postsynaptic proteins, and indeed we see no Syt4-pH accumulation in these structures; however, this does not tell us the relationship between Syt4-pH and the location of a ghost bouton before budding occurs.

Whether Syx4-dependent signaling plays a role in acute activity-dependent plasticity – or, even more interestingly, in instructing budding in a localized fashion – is an open question that we are still actively addressing. As argued in our Discussion, the data in this paper are also consistent with a requirement for Syx4, Syt4, and Nlg1 throughout development to potentiate the synapse.

Author response image 1.NMJ from an animal expressing Syt4-pH postsynaptically (green).The postsynaptic scaffold is labeled with Dlg (red). The neuronal membrane is labeled with HRP (blue). A single ghost bouton is indicated with an arrow. The ghost bouton is devoid of Syt4-pH signal, and there is no obvious change in Syt4-pH at or near the bouton from which the ghost bouton emerged.**DOI:**
http://dx.doi.org/10.7554/eLife.13881.025

Analyses requiring any manual or subjective steps (e.g. bouton number, defining ROI) must be performed blind to genotype. I could not find this information in the paper. If this has not been performed blind to genotype, the data should be reanalysed.

We appreciate this important critique. In the original submitted manuscript, active zone counting analyses were performed blind, but bouton counting was not. We have now repeated all data analyses blind to genotype.

Analyses must be based on an adequate number of independent experiments. I could not find this information in the Materials and methods. For example, the bouton analysis is often stated n=8, n=12, etc. For independent experiments this is a reasonable number. However, if all 8 or 12 larvae came from the same vial (or even couple of vials), this is not acceptable – then there is no way to distinguish an effect of vial from an effect of genotype.

This information has been added to the Methods section.

In the subsection “Nlg mobility is not affected by loss of Syntaxin 4” and Figure 6. The authors use M-W (U?) tests with n=4. This is a non-parametric test that considers only rank order and throws away quantitative data, making it not very effective at detecting significant effects. I don't see how n of 4 can be high enough to give such significant P values, and conversely, a non-significant P value can result from the low statistical power (and n) of this test. Even for a parametric test, an n of 4 is barely acceptable. The authors need a better way to interpret their data and draw robust conclusions, e.g. increase n, and/or compare a parameter that is normally distributed.

We have repeated these experiments and have increased the sample size. We now perform Student’s t test to test for statistically significant differences between samples. This information is included in [Supplementary-material SD4-data].